# Factors affecting revisit intention for medical services at dental clinics

Sewon Park[1], Han-Kyoul Kim[2,3], Mankyu Choi[4,5‡]*, Munjae Lee[6,7‡]*

1 Department of Medical Device Management and Research, SAIHST, Sungkyunkwan University, Seoul, South Korea, 2 Department of Rehabilitation Medicine, Seoul National University Hospital, Seoul, South Korea, 3 National Traffic Injury Rehabilitation Research Institute, National Traffic Injury Rehabilitation Hospital, Yang-Pyeong, South Korea, 4 BK21 FOUR R&E Center for Learning Health Systems, Korea University, Seoul, South Korea, 5 Department of Health Policy & Management, College of Health Science, Korea University, Seoul, South Korea, 6 Department of Medical Humanities and Social Medicine, Ajou University School of Medicine, Suwon, South Korea, 7 Medical Research Collaborating Center, Ajou Research Institute for Innovative Medicine, Ajou University Medical Center, Suwon, South Korea

☯ These authors contributed equally to this work.
‡ These authors also contributed equally to this work.
* mkchoi@korea.ac.kr (MC); emunjae@ajou.ac.kr (ML)

**Data Availability Statement:** All relevant data are within the manuscript and its Supporting Information files. S1 Appendix. Questionnaire for study participants https://doi.org/10.6084/m9.figshare.14418620.v1

## Abstract

Recent changes in the medical paradigm highlight the importance of patient-centered communication. However, because of the lack of awareness of dental clinics and competency of medical personnel, the quality of medical services in terms of the communication between doctors and patients has not improved. This study analyzed the impact of health communication and medical service quality, service value, and patient satisfaction on the intention to revisit dental clinics. The study participants were outpatients treated at 10 dental clinics in Seoul. The research data were collected using a questionnaire during visits to these dental clinics from December 1 to December 30, 2016. A total of 600 questionnaires were distributed (60 copies to each clinics) and 570 valid questionnaires were used for the analysis. The influence of the factors was determined using structural equation modeling. The factors influencing service value were reliability ($\beta = 0.364$, $p < 0.001$), expertise ($\beta = 0.319$, $p < 0.001$), communication by doctors ($\beta = 0.224$, $p < 0.001$), and tangibility ($\beta = 0.136$, $p < 0.05$). In addition, the factors influencing patient satisfaction were reliability ($\beta = 0.258$, $p < 0.001$), tangibility ($\beta = 0.192$, $p < 0.001$), communication by doctors ($\beta = 0.163$, $p < 0.001$), and expertise ($\beta = 0.122$, $p < 0.01$). Further, service value ($\beta = 0.438$, $p < 0.001$) raised patient satisfaction, which was found to influence the intention to revisit dental clinics ($\beta = 0.383$, $p < 0.001$). Providing accurate medical services to inpatients based on smooth communication between doctors and patients improves patient satisfaction. In addition, doctors can build long-term relations with patients by increasing patients' intention to revisit through patient-oriented communication.

**Funding:** This work was supported by the Ministry of Education of the Republic of Korea and the National Research Foundation of Korea (NRF-2019S1A5A2A03040304).

**Competing interests:** The authors have declared that no competing interests exist.

## Introduction

Medical services are changing from a disease-centered model to a patient-centered model. In the existing disease-centered model, all decisions on patient care are made based on the expertise of doctors and other medical personnel. However, in the patient-centered model, patients actively participate in their treatment process and their needs and preferences are reflected in care-related decision making [1, 2]. These changes in the decision-making structure of medical services create competition among medical institutions, forcing them to take steps to survive financially. For instance, they are employing strategies to understand and satisfy patients' needs, much like general commercial enterprises.

In general, to receive care in Korea, primary and secondary medical institutions must be visited first. Subsequently, patients with major ailments are issued with a medical referral form, and care can be received at a tertiary medical institution. Most dental clinics are categorized as primary and secondary medical institutions, and primary medical institutions can receive treatment at the tertiary medical institutions. In addition, most dental clinics tend to have outpatients and relatively few patients with severe diseases. Therefore, it is necessary to increase patients' revisit intention through patient-centered communication so that they can choose the dental clinic in which they would want to receive their treatment [3].

Appropriate communication between doctors and patients provides the latter with information about their treatment based on empathy and understanding and goes beyond mere communication. It may also ensure effective healthcare by enabling joint decision making between physicians and patients [4]. Therefore, healthcare providers should offer their services in a patient-oriented manner. Further, patient-centered communication reduces medical expenditure by decreasing the possibility of unnecessary testing [5].

Healthcare service quality refers to medical services that maximize the welfare of patients while balancing the expected benefits and losses during the treatment process [6]. Healthcare service quality meets the needs of the patient based on the service outcome, service process, and physical environment. Patients tend to have difficulty in evaluating service quality before experiencing it in person. Indeed, even when a service is provided, it is difficult to assess its quality unless a specific problem occurs [7].

Patient satisfaction is a continuous value judgment based on the stimuli associated with the periods before and after consumers' use of medical services. Patients assess medical services based on their own standards, judge the value of those services, and provide a certain response. The result of medical consumers' evaluation can influence their revisit and positive word-of-mouth behavior, which affects the profitability of medical institutions markedly. Service value affects consumer satisfaction through the exchange of cost, time, and service quality. In particular, medical institutions must understand the value of the medical services offered to patients. Medical service value is a concept used to describe or predict the response of a medical consumer; service production itself does not refer to an inherent value but to several parts such as perceived service quality, which form the total service value [8]. Medical institutions should increase consumers' revisit intention by improving the quality of medical services. As such, the importance of the quality of medical services as perceived by patients is emphasized through the provision of patient-centered medical services, which aim to raise patient satisfaction and the revisit intention for healthcare services.

Recently, the concepts of service quality, patient satisfaction, and the relationship between revisit intention and service value have been considered. Prior studies have mainly examined the extent to which service quality affects both satisfaction and revisit intention as well as how satisfaction affects revisit intention. However, most deal with medical service quality, patient satisfaction, and revisit intention individually, and few studies analyze the relationships among them [9–11].

This study analyzed the effect of health communication and service quality on service value, patient satisfaction, and revisit intention, focusing on dental clinics with a high number of patient interactions. To this end, the null hypothesis is that health communication and medical service quality do not affect the revisit intention of dental clinics through the mediating effects of service value and patient satisfaction. Dental services mainly comprise treatment for caries, implants, orthodontics, and oral care. Therefore, since patients require continuous management, they tend to continue to receive treatment in the clinics in which they were first treated. Therefore, for dental clinics, it is important to identify the factors affecting patients' revisit intention. This may help improve the competitiveness of primary medical institutions in Korea by analyzing the correlations among health communication quality, medical service quality, patient satisfaction, service value, and revisit intention.

## Materials and methods

### Research model

Fig 1 shows the research model. This study analyzed the structural relationship between health communication quality and the intention to revisit medical institutions through patient satisfaction and service value. For this, health communication and medical service quality were used as independent variables and revisit intention was used as the dependent variable. Furthermore, patient satisfaction and service value were used as parameters. Communication by doctors and assistants were selected as sub-items of health communication quality and expertise, reliability, tangibility, and accessibility were selected as sub-items of medical service quality.

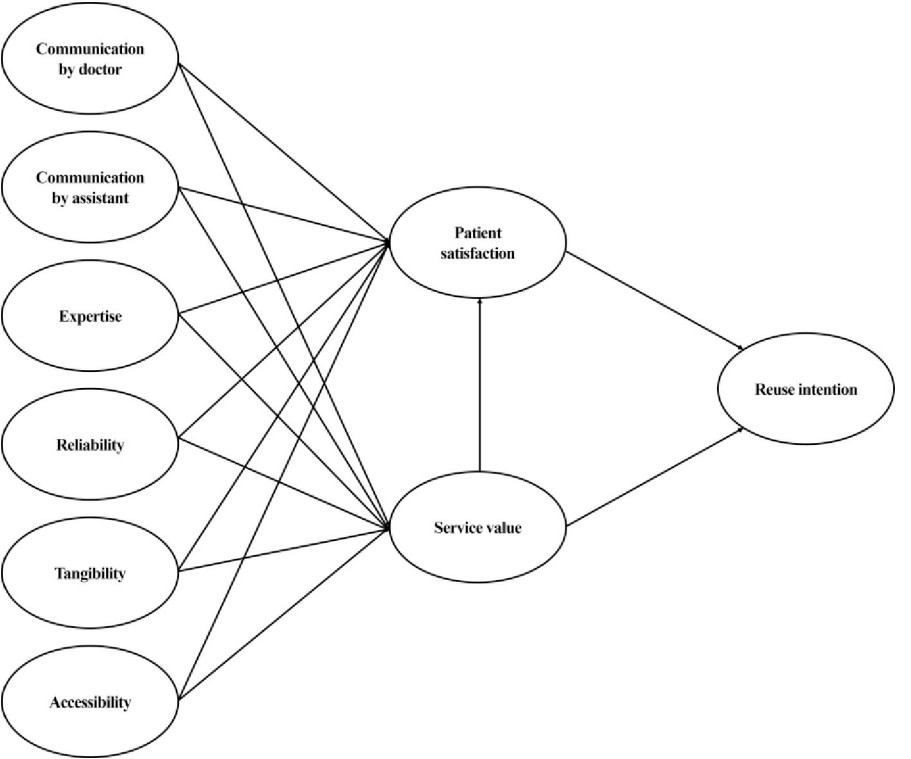

**Fig 1. Research model.**

## Data source and research participants

The study population consisted of outpatients at dental clinics in Seoul. Data were collected using questionnaires during dental clinic visits in Seoul from December 1 to December 30, 2016 (S1 Appendix). Since research measuring the quality of health communication and medical services for dental clinics is scarce, it is expected that these data will help formulate a plan to increase dental clinic revisit intention. First, we determined that there were 941 dental clinics in Seoul from the Korean Medical Practitioners Association. Next, we selected 15 dental clinics using stratified random sampling. Since cooperation with the clinic director was necessary for this research, the official research cooperation document was sent to the director one week before the study commenced using the contact information of dental clinics provided by the Korean Medical Practitioners Association. To confirm directors' cooperation in advance, we recontacted them two days before the survey and informed them of the institution to which the researcher belonged, research purpose, and visit date. Through this process, we obtained cooperation from 10 dental clinics.

We focused on patients aged over 13 years waiting to make a payment or receive their prescription after receiving treatment as an outpatient in the dental clinic. Since oral care can lead to chronic diseases, regular checkups are required from adolescence. The quality of dental services received by patients during adolescence can become an obstacle to continuous visits to dental institutions in adulthood; therefore, adolescent patients were included in the study. The survey method involved researchers and trained investigators informing patients that they belonged to external research institutes, briefly explaining the purpose of the study and content of the questionnaire, and distributing the questionnaires. The self-administered questionnaire was collected immediately after the patient had completed it. A total of 600 questionnaires were distributed and collected, with 60 copies in each of the 10 dental clinics. However, there were 30 incomplete questionnaires because of the short waiting time for payment and receiving the prescription. Of the 600 collected copies, 570 valid copies were thus used for the analysis. The study did not include participants in vulnerable environments or collect or record personally identifiable information. We also did not collect or record sensitive information in accordance with Article 23 of the Privacy Act.

## Research tool

Communication is not only the exchange of information or transmission of opinions, but also the conveyance and understanding of meanings and exchange of emotions. This study constructed a questionnaire based on the measurement items developed by Bowers et al. [12], Marley et al. [13], and Goleman [14]. The questionnaire items for communication by doctors and assistants were revised to reflect communication between doctors and patients; the measurement was conducted using five items each for doctors and assistants [15, 16].

Concerning medical service quality, the SERVOPERF measurement model of Cronin and Taylor was utilized [17, 18]. To measure service quality using the SERVPERF model, 17 questions were used: three items for expertise, four for reliability, six for tangibility, and four for accessibility [19–21].

Patient satisfaction was measured using the measurement items developed by Westbrook [22], Woodside et al. [23], and Dodd et al. [24]. Since higher patient satisfaction in dental clinics indicates high revisit intention more than in other medical institutions, some modifications were made to reflect the characteristics of dental clinics in Korea [25].

Service value is the physical and emotional value the patient experiences through the treatment process and results. Some modifications were made to consider the characteristics of Korean dental clinics based on the measurement items developed by Gooding and Cronin et al. [26–28].

Revisit intention refers to the intentions of healthcare users to maintain a continuing transaction with a healthcare provider after experiencing its services. The measurement items developed by Swan and Reidenbach and Sandifer-Smallwood were utilized. As the word-of-mouth effect of existing patients can significantly influence choosing a medical institution, especially a dental clinic, some modifications were made to reflect the characteristics of dental services [29–32].

## Data analysis

Data analysis was conducted using SPSS 25.0 (IBM, Chicago, IL, USA) and Amos 18.0 (IBM, Chicago, IL, USA) software. The specific analysis method was as follows. First, a frequency analysis was conducted to determine the demographic characteristics of the participants. Second, a factor analysis was performed to verify the validity of the questions, while the reliability of the measurement questions was validated using Cronbach's α. For the factor analysis, an exploratory factor analysis (EFA) of the Varimax mode orthogonal rotation was first performed to examine the factor structure of the questions to measure the variables. Next, a confirmatory factor analysis (CFA) was conducted to confirm whether the derived factor structure was consistent with the actual empirical data. Third, structural equation modeling was utilized to analyze the structural relationships influencing each factor. The structural equation models were analyzed using a two-step approach. First, a CFA was conducted on the individual measurement models or simultaneously on the factors and variables included in both the measurement model and the theoretical model. This process confirmed the reliability in a single dimension and the validity between concepts. Second, we linked and analyzed the factors that appeared in the research model and evaluated the structural relationships.

## Results

### Demographics

Participants' age range showed an even distribution, with 171 people (30%) in their 20s being the largest age group. Concerning academic background, the largest percentage—361 people —had graduated from college (63.3%); for average income, 158 people had a monthly income of more than 5 million won (27.7%). Regarding the question of whether it was their first medical examination, 88.6% of respondents answered that it was not; on the reason for visiting the dental clinic question, cavity treatment accounted for the largest group (33.5%), followed by scaling (14%), and endodontic treatment (12.6%). For the time spent visiting the dental clinic, approximately 77.7% responded that it was less than 30 minutes, and for their reason for visiting, accessibility (32.3%) accounted for the largest group, followed by excellent medical staff (22.6%) and referral by acquaintances (15.6%). This may be because dental treatment often requires continuous treatment and people tend to use accessible dental clinics for an extended period since a large number of patients visit the clinic to receive regular checkups (Table 1).

### Reliability and study model verification

An EFA was conducted based on the collected data to examine the factor structure of the 48 questions used to measure the variables. For exploratory factor analysis, the validity of the composition was verified using the principal components analysis (PCA) of the Varimax rotation, and Kaise-Meyer Olkin (KMO) and Barlett sphericity were verified. Variables were selected based on an eigenvalue of 1 or more and factor loading of 0.4 or more for each variable, and Cronbach's Alpha was checked for reliability, and items that lowered reliability were removed through factor analysis and improved to an appropriate level. As a result, six items

**Table 1. Participants' demographic characteristics and medical utilization behavior.**

| Type | | No. | % |
|---|---|---|---|
| Sex | Male | 231 | 41.0 |
| | Female | 339 | 59.0 |
| Age | Under 20 years | 23 | 4.0 |
| | 20 to 29 years | 171 | 30.0 |
| | 30 to 39 years | 114 | 20.0 |
| | 40 to 49 years | 114 | 20.0 |
| | 50 to 59 years | 86 | 15.1 |
| | 60 years or over | 62 | 10.9 |
| Education | Under middle school | 44 | 7.7 |
| | High school | 165 | 29.0 |
| | Junior college | 312 | 54.7 |
| | University and above | 49 | 8.6 |
| Income | Under 1 million won | 13 | 2.3 |
| | 1–2 million won | 48 | 8.4 |
| | 2–3 million won | 93 | 16.3 |
| | 3–4 million won | 125 | 21.9 |
| | 4–5 million won | 133 | 23.3 |
| | 5 million won and over | 158 | 27.7 |
| First visit | First visit | 65 | 11.4 |
| | Returning patients | 505 | 88.6 |
| Reason for visit | Cavity treatment | 191 | 33.5 |
| | Endodontic treatment | 72 | 12.6 |
| | Implant | 59 | 10.4 |
| | Scaling | 80 | 14.0 |
| | Ache | 23 | 4.0 |
| | Checkup | 15 | 2.6 |
| | Whitening | 10 | 1.8 |
| | Correction | 53 | 9.3 |
| | Prosthetic treatment | 29 | 5.1 |
| | Denture | 11 | 1.9 |
| | Gum treatment | 21 | 3.7 |
| | Other | 6 | 1.1 |
| Time required for visit | Within 15 minutes | 306 | 53.7 |
| | Within half an hour | 137 | 24.0 |
| | Within an hour | 78 | 13.7 |
| | Within an hour and a half | 49 | 8.6 |
| Reason for selection | Accessibility | 184 | 32.3 |
| | Referral by acquaintances | 89 | 15.6 |
| | Cheap medical expenses | 29 | 5.1 |
| | Excellent medical staff | 129 | 22.6 |
| | Convenient medical procedure | 15 | 2.6 |
| | Kindness | 29 | 5.1 |
| | Sanitary condition | 10 | 1.8 |
| | Convenient hospital facilities | 11 | 1.9 |
| | Hospital reputation | 30 | 5.3 |
| | Other | 44 | 7.7 |

(*Continued*)

**Table 1.** (Continued)

| Type | | No. | % |
|---|---|---|---|
| Selection method | Perimeter solicitation | 319 | 56.0 |
| | Internet | 70 | 12.3 |
| | Advertisement | 36 | 6.3 |
| | Hospital recommendation | 9 | 1.6 |
| | Sign | 105 | 18.4 |
| | Homepage | 6 | 1.0 |
| | Other | 25 | 4.4 |

including the expertise of assistants and responsiveness of the office/clinic had commonality less than 0.4 and were deleted. Hence, 42 questions were finally selected. The EFA was again conducted to examine the factor structure of the final selected items. The Kaiser–Meyer–Olkin test value was 0.944, while Bartlett's test of sphericity was also significant ($\chi_2$ = 13748.522, p < 0.001). Therefore, the data used in the analysis were judged as suitable for the factor analysis. In addition, the total variance explained was 74% (Table 2).

The Cronbach's α values were 0.905 for communication by doctors, 0.932 for communication by assistants, 0.897 for expertise, 0.928 for reliability, 0.887 for tangibility, 0.820 for accessibility, 0.959 for patient satisfaction, 0.938 for service value, and 0.970 for revisit intention. The EFA was classified into the remaining nine factors. Hence, the Cronbach's α values of the variables used in the study were very high (Table 3).

To verify the internal validity of the model, a CFA was conducted on the questions of the measurement model. To evaluate the appropriateness of the CFA, the $\chi_2$ value, p-value for $\chi_2$ value, Tucker–Lewis index (TLI), comparative fit index (CFI), and root mean square error of approximation (RMSEA) were used. The coefficient values were estimated as $\chi_2$ = 1712.643 (df = 783, p < 0.001), TLI = 0.918, and CFI = 0.926, suggesting that the model fit was excellent overall. In addition, RMSEA = 0.063 was less than 0.08, making the factor analysis reasonable. In the CFA, two items for communication by doctors, one for tangibility, and one for accessibility did not exceed the 0.5 standardized factor loading criterion. Hence, of the 48 questions used to collect the data, six items with poor commonality were removed through the EFA and four items with poor validity were removed through the CFA. Therefore, 38 items were used for the analysis. Table 4 shows the results of the CFA for the model used in this study.

## Structural equation model verification

The results of analyzing the model used in the study showed that $\chi_2$ = 1653.662, TLI = 0.910, CFI = 0.917, and RMSEA = 0.072, indicating that the values of the indexes were generally excellent. Table 5 shows the goodness-of-fit of the research model.

Table 6 shows the standardized path coefficient values and significance levels. First, the quality factors of dental services that affect service value included communication by doctors, expertise, reliability, and tangibility, all of which were found to have a positive impact. Investigating the influence of each factor separately, reliability (0.364) showed the highest influence on service value, followed by expertise (0.319) and communication by doctors (0.224). Conversely, communication by assistants and accessibility did not affect service value.

Next, the quality factors that affect patient satisfaction included communication by doctors, expertise, reliability, and tangibility, all of which were found to have a positive impact. Specifically, the influence of reliability (0.258) was the highest on patient satisfaction, followed by tangibility (0.192) and communication by doctors (0.163). Furthermore, similar to service value,

**Table 2. EFA results.**

| Variable | | Commonality | Component | | | | | | | | |
|---|---|---|---|---|---|---|---|---|---|---|---|
| | | | 1 | 2 | 3 | 4 | 5 | 6 | 7 | 8 | 9 |
| Communication by doctor | Doctor1 | 0.698 | 0.295 | | | | | | | | |
| | Doctor2 | 0.766 | 0.292 | | | | | | | | |
| | Doctor3 | 0.750 | 0.251 | | | | | | | | |
| | Doctor4 | 0.769 | 0.208 | | | | | | | | |
| | Doctor5 | 0.794 | 0.184 | | | | | | | | |
| | Doctor6 | 0.749 | 0.182 | | | | | | | | |
| | Doctor7 | 0.766 | 0.136 | | | | | | | | |
| Communication by assistant | Assistant1 | 0.755 | | 0.818 | | | | | | | |
| | Assistant2 | 0.828 | | 0.798 | | | | | | | |
| | Assistant3 | 0.814 | | 0.791 | | | | | | | |
| | Assistant4 | 0.808 | | 0.751 | | | | | | | |
| | Assistant5 | 0.823 | | 0.732 | | | | | | | |
| Expertise | Expertise1 | 0.672 | | | 0.695 | | | | | | |
| | Expertise2 | 0.754 | | | 0.661 | | | | | | |
| | Expertise3 | 0.773 | | | 0.603 | | | | | | |
| Expertise of assistant staff | Expertise of assistant1 | 0.312 | | | | | | | | | |
| | Expertise of assistant2 | 0.339 | | | | | | | | | |
| | Expertise of assistant3 | 0.392 | | | | | | | | | |
| Reliability | Reliability1 | 0.789 | | | | 0.606 | | | | | |
| | Reliability2 | 0.785 | | | | 0.571 | | | | | |
| | Reliability3 | 0.725 | | | | 0.566 | | | | | |
| | Reliability4 | 0.719 | | | | 0.525 | | | | | |
| responsiveness | Responsiveness1 | 0.234 | | | | | | | | | |
| | Responsiveness2 | 0.351 | | | | | | | | | |
| | Responsiveness3 | 0.256 | | | | | | | | | |
| Tangibility | Tangibility1 | 0.692 | | | | | 0.847 | | | | |
| | Tangibility2 | 0.692 | | | | | 0.830 | | | | |
| | Tangibility3 | 0.679 | | | | | 0.806 | | | | |
| | Tangibility4 | 0.814 | | | | | 0.758 | | | | |
| | Tangibility5 | 0.809 | | | | | 0.646 | | | | |
| | Tangibility6 | 0.790 | | | | | 0.495 | | | | |
| Accessibility | Accessibility1 | 0.804 | | | | | | 0.817 | | | |
| | Accessibility2 | 0.814 | | | | | | 0.786 | | | |
| | Accessibility3 | 0.721 | | | | | | 0.775 | | | |
| | Accessibility4 | 0.757 | | | | | | 0.673 | | | |
| Patient satisfaction | Patient satisfaction1 | 0.810 | | | | | | | 0.655 | | |
| | Patient satisfaction2 | 0.825 | | | | | | | 0.640 | | |
| | Patient satisfaction3 | 0.846 | | | | | | | 0.638 | | |
| | Patient satisfaction4 | 0.798 | | | | | | | 0.568 | | |
| Service value | Service value1 | 0.855 | | | | | | | | 0.816 | |
| | Service value2 | 0.850 | | | | | | | | 0.800 | |
| | Service value3 | 0.799 | | | | | | | | 0.797 | |
| | Service value4 | 0.773 | | | | | | | | 0.785 | |
| | Service value5 | 0.795 | | | | | | | | 0.726 | |

(*Continued*)

**Table 2.** (Continued)

| Variable | | Commonality | Component | | | | | | | | |
|---|---|---|---|---|---|---|---|---|---|---|---|
| | | | 1 | 2 | 3 | 4 | 5 | 6 | 7 | 8 | 9 |
| Revisit intention | Revisit intention1 | 0.882 | | | | | | | | | 0.685 |
| | Revisit intention2 | 0.915 | | | | | | | | | 0.631 |
| | Revisit intention3 | 0.906 | | | | | | | | | 0.626 |
| | Revisit intention4 | 0.899 | | | | | | | | | 0.593 |
| Eigenvalue | | | 21.08 | 3.29 | 2.65 | 1.96 | 1.83 | 1.41 | 1.25 | 1.08 | 1.02 |
| Explained variance (%) | | | 16.2 | 11.13 | 9.83 | 9.75 | 9.53 | 6.15 | 5.03 | 3.49 | 2.89 |
| Total explained variance (%) | | | 16.26 | 27.39 | 37.21 | 46.96 | 56.54 | 62.69 | 67.71 | 71.21 | 74.10 |

communication by assistants and accessibility did not affect patient satisfaction. Additionally, service value (0.444)—the endogenous variable—also raised patient satisfaction. Fig 2 shows the influence of each factor.

## Discussion

The present study analyzed the effect of the quality of health communication and medical services on service value, patient satisfaction, and revisit intention. We found that the quality of health communication and medical services influenced the revisit intention of dental clinics through the mediating effects of patient satisfaction and service value, thus rejecting the null hypothesis and accepting the alternative hypothesis. The detailed results are as follows.

First, reliability and communication by doctors raised patient satisfaction and service value. These results are similar to those of Chang et al. (2013), who found that the doctor's communication attitude affects patient satisfaction, medical service quality, and reliability [33]. In addition, Rashid et al. (2014) reported that communication by doctors raises patients satisfaction more than clinical competency [34]. Further, it has been found that the empathy of hospital staff, a form of communication, affects patient satisfaction and revisit intention markedly [10, 35, 36]. Since outpatients—unlike inpatients—need to be provided with only the necessary medical services, factors such as convenient treatment, administrative procedures, and the kindness of medical staff may influence their satisfaction. In particular, dental clinics, which only see outpatients, can obtain accurate information on patients and provide the necessary medical services by utilizing doctors' communication skills. This leads to a consistent increase in the reliability of healthcare services, resulting in improved service value and patient satisfaction.

**Table 3. Reliability verification.**

| Variable | No. of items | Construct reliability (Cronbach's α) |
|---|---|---|
| Communication by doctors | 7 | 0.905 |
| Communication by assistants | 5 | 0.932 |
| Expertise | 3 | 0.897 |
| Reliability | 4 | 0.928 |
| Tangibility | 6 | 0.887 |
| Accessibility | 4 | 0.820 |
| Patient satisfaction | 4 | 0.959 |
| Service value | 5 | 0.938 |
| Revisit intention | 4 | 0.970 |

**Table 4. CFA results.**

| Factor | Path | Estimate | S.E. | T | p-value | Standardized estimate | SMC |
|---|---|---|---|---|---|---|---|
| Health communication quality | Doctor1 ← Health communication | 1.000 | | | | 0.773 | 0.599 |
| | Doctor2 ← Health communication | 1.183 | 0.074 | 16.024 | 0.001 | 0.852 | 0.725 |
| | Doctor3 ← Health communication | 1.009 | 0.067 | 15.145 | 0.001 | 0.814 | 0.660 |
| | Doctor4 ← Health communication | 1.099 | 0.072 | 15.298 | 0.001 | 0.821 | 0.674 |
| | Doctor5 ← Health communication | 1.129 | 0.071 | 15.811 | 0.001 | 0.843 | 0.714 |
| | Doctor6 ← Health communication | 0.872 | 0.081 | 10.783 | 0.001 | 0.372 | 0.610 |
| | Doctor7 ← Health communication | 0.886 | 0.082 | 10.772 | 0.001 | 0.371 | 0.609 |
| | Assistant1 ← Health communication | 1.000 | | | | 0.832 | 0.754 |
| | Assistant2 ← Health communication | 0.980 | 0.054 | 18.112 | 0.001 | 0.850 | 0.743 |
| | Assistant3 ← Health communication | 0.999 | 0.053 | 18.931 | 0.001 | 0.874 | 0.763 |
| | Assistant4 ← Health communication | 1.049 | 0.057 | 18.532 | 0.001 | 0.862 | 0.722 |
| | Assistant5 ← Health communication | 1.044 | 0.056 | 18.743 | 0.001 | 0.868 | 0.692 |
| Expertise | Expertise1 ← Expertise | 1.000 | | | | 0.839 | 0.703 |
| | Expertise2 ← Expertise | 0.972 | 0.054 | 18.004 | 0.001 | 0.857 | 0.734 |
| | Expertise3 ← Expertise | 1.138 | 0.055 | 18.952 | 0.001 | 0.889 | 0.791 |
| Reliability | Reliability1 ← Reliability | 1.000 | | | | 0.896 | 0.802 |
| | Reliability2 ← Reliability | 1.059 | 0.043 | 24.686 | 0.001 | 0.918 | 0.843 |
| | Reliability3 ← Reliability | 1.024 | 0.051 | 20.198 | 0.001 | 0.838 | 0.702 |
| | Reliability4 ← Reliability | 1.046 | 0.051 | 20.312 | 0.001 | 0.840 | 0.706 |
| Tangibility | Tangibility1 ← Tangibility | 1.000 | | | | 0.754 | 0.568 |
| | Tangibility2 ← Tangibility | 1.153 | 0.085 | 13.611 | 0.001 | 0.765 | 0.585 |
| | Tangibility3 ← Tangibility | 1.227 | 0.077 | 15.998 | 0.001 | 0.882 | 0.778 |
| | Tangibility4 ← Tangibility | 1.239 | 0.078 | 15.955 | 0.001 | 0.880 | 0.774 |
| | Tangibility5 ← Tangibility | 1.306 | 0.086 | 15.130 | 0.001 | 0.839 | 0.705 |
| | Tangibility6 ← Tangibility | 0.979 | 0.110 | 8.924 | 0.001 | 0.271 | 0.520 |
| Accessibility | Accessibility1 ← Accessibility | 1.000 | | | | 0.902 | 0.813 |
| | Accessibility2 ← Accessibility | 1.083 | 0.053 | 20.472 | 0.001 | 0.934 | 0.872 |
| | Accessibility3 ← Accessibility | 0.874 | 0.067 | 13.048 | 0.001 | 0.656 | 0.430 |
| | Accessibility4 ← Accessibility | 0.605 | 0.081 | 7.493 | 0.001 | 0.271 | 0.520 |
| Service value | Service1 ← Service value | 1.000 | | | | 0.941 | 0.885 |
| | Service2 ← Service value | 1.014 | 0.034 | 30.273 | 0.001 | 0.927 | 0.859 |
| | Service3 ← Service value | 0.907 | 0.039 | 22.974 | 0.001 | 0.845 | 0.715 |
| | Service4 ← Service value | 0.801 | 0.045 | 17.679 | 0.001 | 0.751 | 0.564 |
| | Service5 ← Service value | 0.932 | 0.039 | 23.868 | 0.001 | 0.858 | 0.736 |
| Patient satisfaction | Satisfaction1 ← Satisfaction | 1.000 | | | | 0.908 | 0.825 |
| | Satisfaction2 ← Satisfaction | 1.025 | 0.035 | 28.949 | 0.001 | 0.942 | 0.888 |
| | Satisfaction3 ← Satisfaction | 1.080 | 0.036 | 29.909 | 0.001 | 0.952 | 0.907 |
| | Satisfaction4 ← Satisfaction | 1.057 | 0.042 | 25.243 | 0.001 | 0.899 | 0.809 |
| Revisit intention | Revisit1 ← Revisit intention | 1.000 | | | | 0.944 | 0.892 |
| | Revisit2 ← Revisit intention | 1.023 | 0.026 | 39.650 | 0.001 | 0.972 | 0.945 |
| | Revisit3 ← Revisit intention | 0.968 | 0.028 | 34.935 | 0.001 | 0.947 | 0.897 |
| | Revisit4 ← Revisit intention | 0.978 | 0.033 | 29.875 | 0.001 | 0.912 | 0.832 |

***p<0.001, S.E. = Standard error, T = t-value, β = Standardized coefficient, SMC = Squared multiple correlation.

Second, the results indicated that communication by assistants did not affect patient satisfaction or service value in contrast to previous research results. Ehsan et al. (2015) showed that

**Table 5. Research model verification (N = 570).**

|  | $\chi^2$ | DF | p-value | TLI | CFI | RMSEA |
|---|---|---|---|---|---|---|
| Research model | 1653.662 | 644 | 0.001 | 0.910 | 0.917 | 0.072 |

***p<0.001, $X^2$ = Chi-square statistic, DF = Degrees of freedom, TLI = Tucker–Lewis index, CFI = Comparative fit index, RMSEA = Root mean square error of approximation.

smoother communication between doctors and assistants raises patient satisfaction [37]. In addition, Fellani Danasra et al. (2011) reported that most patients receiving dental treatment wish to communicate with assistants about their discomfort in treatment, which subsequently affects the patient's intention to revisit dental clinics [38]. Because dental clinics have a longer period of medical treatment than general medical institutions, doctors' communication with patients is more important than that by assistants. Therefore, to improve service value and patient satisfaction in dental clinics, patient-centered communication by doctors is required. In other words, doctors should understand and respect the position of patients and recognize the importance of communication skills, focusing on providing a sufficient explanation and conveying an expert knowledge of the treatment.

Third, patient satisfaction and service value influenced the intention to revisit dental clinics. According to Seema (2011), patient satisfaction improves compliance with the treatment process and helps to maintain treatment continuity, thereby influencing the revisit intention of a medical institution [39]. In addition, Anang et al. (2019) showed that service quality at a medical institution affects patient satisfaction [40]. Further, previous research has found significant correlations among outpatients satisfaction, service quality, and revisit intention [40–42]. Patients visit the medical institution that meets their selection criteria and continue to be provided with medical services from that medical institution. In particular, dental treatments such as caries, implants, and orthodontics usually take two to three years, rather than being one-off

**Table 6. Research model path coefficients.**

| Factor | Path | B | β | S.E. | T | p-value |
|---|---|---|---|---|---|---|
| Service value | Service ← Communication by doctors | 0.215 | 0.224 | 0.060 | 3.600** | 0.001 |
|  | Service ← Communication by assistants | -0.038 | -0.037 | 0.066 | -0.580 | 0.562 |
|  | Service ← Expertise | 0.321 | 0.319 | 0.086 | 3.748*** | 0.001 |
|  | Service ← Reliability | 0.365 | 0.364 | 0.089 | 4.113*** | 0.001 |
|  | Service ← Tangibility | 0.175 | 0.136 | 0.081 | 2.145** | 0.032 |
|  | Service ← Accessibility | 0.014 | 0.014 | 0.052 | 0.259 | 0.795 |
| Patient satisfaction | Satisfaction ← Communication by doctors | 0.140 | 0.163 | 0.044 | 3.211** | 0.001 |
|  | Satisfaction ← Communication by assistants | 0.046 | 0.050 | 0.046 | 0.997 | 0.319 |
|  | Satisfaction ← Expertise | 0.110 | 0.122 | 0.062 | 1.778* | 0.075 |
|  | Satisfaction ← Reliability | 0.231 | 0.258 | 0.064 | 3.585*** | 0.001 |
|  | Satisfaction ← Tangibility | 0.220 | 0.192 | 0.059 | 3.729*** | 0.001 |
|  | Satisfaction ← Accessibility | -0.005 | -0.005 | 0.037 | -0.130 | 0.897 |
|  | Satisfaction ← Service value | 0.397 | 0.444 | 0.050 | 7.941*** | 0.001 |
| Revisit intention | Revisit ← Patient satisfaction | 0.491 | 0.383 | 0.087 | 5.616*** | 0.001 |
|  | Revisit ← Service value | 0.501 | 0.438 | 0.078 | 6.414*** | 0.001 |

*p<0.1,

**p<0.05,

***p<0.001, B = Unstandardized coefficient, β = Standardized coefficient, S.E. = Standard error, T = t-value

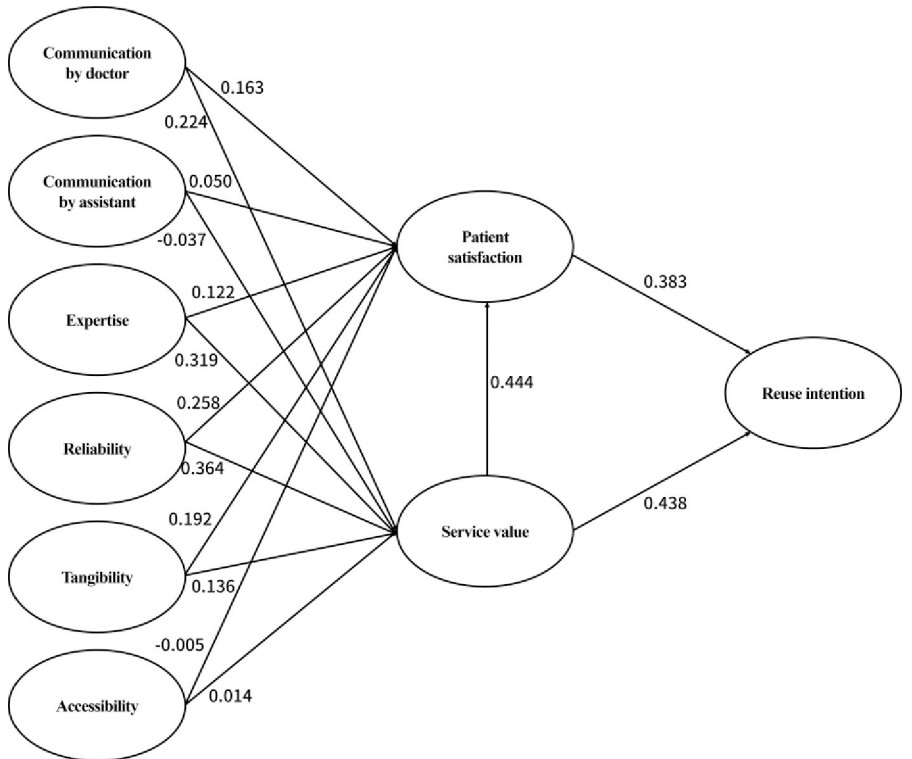

**Fig 2. Final path model.**

treatments; hence, patients have a strong tendency to maintain services in the long run. It is therefore important to retain existing patients by ensuring patient satisfaction and service value. Regarding satisfaction with dental services, the human factor also appears to be more important than in general medical treatment, meaning that the doctor's active communication is again required. Accordingly, providing patient-oriented medical services and strengthening communication between doctors and patients may enhance revisit intention for dental clinics by improving service value and patient satisfaction.

Finally, the limitations of this study and future research directions should be noted. First, the questionnaires were distributed and collected from outpatients in selected dental clinics in Seoul. Because the distribution of dental clinics differs regionally, the results of this study cannot easily be generalized to all dental clinics. If the analysis was repeated by including dental clinics in different provinces, accessibility would also affect patient satisfaction and service value. Accordingly, follow-up studies should attempt to overcome this research's regional limitations.

Second, since there are many free-of-charge items in dental treatment, it is necessary to examine their influence on revisit intention for medical services by considering the specificity of dental treatment and including price factors such as medical expenses. In particular, as implant procedures have recently been increasing, a future analysis could be conducted by dividing subjects into beneficiaries and non-beneficiaries of national health insurance to examine the impact of medical expenses on revisit intention. This would help understand the effect of transparent disclosure on medical expenses through communication between doctors and patients and its effect on patient satisfaction.

Third, this study focused on general dental clinics without classifying them into clinics and network hospitals and the structural model was applied to the quality of health communication

and medical services, patient satisfaction, service value, and revisit intention. However, given that the number of network hospitals has recently increased rapidly and the co-branded opening of network hospitals targeting specific patients has become generalized, research should be conducted on image factors, which may have a direct effect on patient satisfaction and service value.

## Conclusion

This study analyzed the factors influencing the intention to revisit medical services using data from patients visiting dental clinics in Seoul. The results showed that reliability and communication by doctors affected service value and patient satisfaction, which influenced revisit intention. The following measures are necessary to increase the satisfaction of patients who visit dental clinics and to increase the revisit intention. Dental clinics should provide appropriate medical services to outpatients based on smooth communication between doctors and patients. Additionally, encouraging doctors to show an attitude of respect toward the patient may affect patient satisfaction. Doctors providing medical treatment information to patients with a friendly and respectful attitude rather than an authoritarian one may be an effective strategy for dental clinics to build long-term relationships with patients.

## Supporting information

**S1 Appendix. Questionnaire for study participants.**
(DOCX)

## Acknowledgments

We are grateful to the journal editors and three anonymous reviewers for taking the time to provide helpful comments to improve the paper.

## Author Contributions

**Conceptualization:** Han-Kyoul Kim, Mankyu Choi.

**Formal analysis:** Sewon Park, Munjae Lee.

**Investigation:** Munjae Lee.

**Methodology:** Sewon Park.

**Project administration:** Mankyu Choi, Munjae Lee.

**Software:** Munjae Lee.

**Supervision:** Mankyu Choi.

**Validation:** Han-Kyoul Kim, Mankyu Choi.

**Writing – original draft:** Sewon Park.

**Writing – review & editing:** Mankyu Choi, Munjae Lee.

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
