## [Decision Letter · Decision Letter 0]

12 Jun 2020

PONE-D-19-29416

Factors affecting revisiting intention for medical services at dental clinics

PLOS ONE

Dear Dr. lee,

Thank you for submitting your manuscript to PLOS ONE. After careful consideration, we feel that it has merit but does not fully meet PLOS ONE’s publication criteria as it currently stands.

Having intensively reviewed your draft, our external referees have indicated major drawbacks. Moreover, our reviewers strongly differed with their final recommendations, and, thus, I have invited a further external referee, to come to a more balanced decision. All in all, the indicated shortcomings are considered reasonable with regard to both PLOS ONE's quality standards and our readership's expectations.

Therefore, we invite you to submit a revised version of the manuscript that addresses all the points raised during the review process. Please note that an insufficient revision not following our reviewers recommendations, or ignoring PLOS ONE's Gidelines for Authors will lead to 'outright reject'.

We look forward to receiving your revised manuscript.

Kind regards,

Andrej M Kielbassa, Prof. Dr. med. dent. Dr. h. c.

Academic Editor

PLOS ONE

Journal Requirements:

The funders had no role in study design, data collection and analysis, decision to

publish, or preparation of the manuscript.

6. Please ensure that you refer to Figure 1 in your text as, if accepted, production will need this reference to link the reader to the figure.

Reviewers' comments:

Reviewer's Responses to Questions

**Comments to the Author**

1. Is the manuscript technically sound, and do the data support the conclusions?

Reviewer #1: No

Reviewer #2: Yes

Reviewer #3: Yes

2. Has the statistical analysis been performed appropriately and rigorously? 

Reviewer #1: Yes

Reviewer #2: Yes

Reviewer #3: Yes

3. Have the authors made all data underlying the findings in their manuscript fully available?

Reviewer #1: No

Reviewer #2: No

Reviewer #3: Yes

4. Is the manuscript presented in an intelligible fashion and written in standard English?

Reviewer #1: Yes

Reviewer #2: No

Reviewer #3: Yes

5. Review Comments to the Author

Reviewer #1: Abstract

- Please provide important information. Presently, this section does contain some self-explaining phrases only. Remember that this part is a stand-alone section, allowing future readers to switch to your main text.

Intro

- Remember to elaborate both aims and objectives more clearly.

- There must be a deducable null hypothesis, reasonable and indisputable.

Meths

- Re your research model, this must be understandable for every reader. Just depicting some kind of chart would not seem appropriate. Please remember to guide the reader.

- Collecting data from December 1 to December 30, 2016 would seem outdated, wouldn't it?

- This section must provide methodology. Do not give a literature review here.

- As always, please provide general names with your text, followed by (brand names; manufacturer, city, country) in parentheses. Please provide full information with ALL materials and methodologies.

Results

- "To test internal consistency among the measurement items, a reliability verification was conducted

using the Cronbach's α value. To measure construct validity, a factor analysis was performed using the

Varimax mode." This obviously refers to methodology. Please revise.

- "Table 3 shows the results of the CFA for the model used in this study." Again, this would not seem sufficient. Do you expect the reader to analyze these data?

Disc

- This would not seem well-elaborated.

- Please discuss outcome, provide insight thoughts on the methodology, and speculate on future research directions. As with the other sections, this part must be re-prganized.

Concl

- Do not simply repeat the results here. This section must provide a reasonable extension of your outcome. Strictly stick to your aims here.

Refs

- Please revise for uniform formatting. Stick to the Guidelines.

In total, this submitted draft would not seem worth following in its present form.

Reviewer #2: This paper describes the factors affecting revisiting intention of dental patients. The topic is very relevant and important. The paper could be enhanced by describing existing knowledge on this topic in the background section and making it more relevant to dentistry. Also there are inconsistencies in calling medical services versus dental clinics is confusing. It is better to stick with dental services and dental clinics if the study was conducted in these settings and to reduce ambiguity. Also, please describe briefly the types of dental care provided in the participating clinics. Finally, the current description the background has a lot of redundant material, which can be shortened.

The methods section again have way too much details and could be difficult for the reader to have a good grasp of the approach. Also, while the figure displaying the research model gives the impression of health communication as an independent variable, it appears health communication is a dependent variable.

The results section could also be improved by organizing results related to the research question and assessing the reliability of the survey questions separately. Also, clarity can be improved by being concise. The discussion section again can be enhanced by describing major findings concisely and clearly. It is also not clear what types of medical services dentists provide. Also, references need to be cited appropriately.

Reviewer #3: The discussion section: lacks referencing as the authors stated several times (previous studies) without giving reference to which study?

For the conclusion section: the limitations should be stated within the discussion section and the conclusion should summarise only the key result and future studies if required.

6. PLOS authors have the option to publish the peer review history of their article (what does this mean?). If published, this will include your full peer review and any attached files.

Reviewer #1: No

Reviewer #2: No

Reviewer #3: No

---

## [Author Response · Author response to Decision Letter 0]

13 Aug 2020

Response to Reviewer Comments

I wish to thank the reviewers for their constructive feedback. The reviewers point out some remaining elements requiring modifications or clarifications in order to validate my manuscript for publication. My manuscript was thoroughly reviewed and updated according to these pertinent remarks. I would like to illustrate these modifications and address those discussion points in my response below.

Point 1: Please ensure that your manuscript meets PLOS ONE's style requirements, including those for file naming. 

Response 1: Has confirmed.

Point 2: Please include additional information regarding the survey or questionnaire used in the study and ensure that you have provided sufficient details that others could replicate the analyses. 

Response 2: 

Questionnaire (S1 Table) added.

Point 3: PLOS requires an ORCID iD for the corresponding author in Editorial Manager on papers submitted after December 6th, 2016. Please ensure that you have an ORCID iD and that it is validated in Editorial Manager.

Response 3: Has confirmed.

Point 4: We note that you have indicated that data from this study are available upon request. PLOS only allows data to be available upon request if there are legal or ethical restrictions on sharing data publicly. 

Response 4: Has confirmed.

Point 5: If there are ethical or legal restrictions on sharing a de-identified data set, please explain them in detail (e.g., data contain potentially identifying or sensitive patient information) and who has imposed them (e.g., an ethics committee). Please also provide contact information for a data access committee, ethics committee, or other institutional body to which data requests may be sent. 

Response 5: 

The study did not include subjects in vulnerable environments and did not collect or record personally identifiable information using information that is disclosed to individuals. We also did not collect or record sensitive information in accordance with Article 23 of the Privacy Act. 

Point 6: If there are no restrictions, please upload the minimal anonymized data set necessary to replicate your study findings as either Supporting Information files or to a stable, public repository and provide us with the relevant URLs, DOIs, or accession numbers. Please see http://www.bmj.com/content/340/bmj.c181.long for guidelines on how to de-identify and prepare clinical data for publication. For a list of acceptable repositories, please see http://journals.plos.org/plosone/s/data-availability#loc-recommended-repositories.

Response 6: 

The authors confirm that all data underlying the findings are fully available without restriction. All relevant data are within the manuscript and its Supporting Information files.

Point 7: Please clarify the sources of funding (financial or material support) for your study. List the grants or organizations that supported your study, including funding received from your institution. 

Response 7: No financial support was provided to fund this manuscript.

Point 8: State what role the funders took in the study. If the funders had no role in your study, please state: “The funders had no role in study design, data collection and analysis, decision to publish, or preparation of the manuscript.”

Response 8:

MC and ML conceptualization, operative project leader. SP and ML performed the statistical analysis, interpreted the data, and helped to draft the manuscript. SP participated in the design of the study and helped to draft the manuscript. ML participated in the data collection and revised manuscript. MC designed the study, was principal investigator, participated in the interpretation of the data and revised the manuscript. All authors have read and approved the final manuscript.

Point 9: If any authors received a salary from any of your funders, please state which authors and which funders. If you did not receive any funding for this study, please state: “The authors received no specific funding for this work.”

Response 9: The authors received no specific funding for this work.

Point 10: Please ensure that you refer to Figure 1 in your text as, if accepted, production will need this reference to link the reader to the figure. 

Response 10: It's corrected.

Point 11: Please include captions for your Supporting Information files at the end of your manuscript, and update any in-text citations to match accordingly. Please see our Supporting Information guidelines for more information: http://journals.plos.org/plosone/s/supporting-information. 

Response 11: It's corrected.

I hope that the considerable changes made to my research paper coupled with our above arguments will convince the reviewers in reconsidering our manuscript for publication in your journal. I thank you in advance for your kind and thorough attention in the review of my work.

Best regards,

Response to Reviewer Comments

I wish to thank the reviewers for their constructive feedback. The reviewers point out some remaining elements requiring modifications or clarifications in order to validate my manuscript for publication. My manuscript was thoroughly reviewed and updated according to these pertinent remarks. I would like to illustrate these modifications and address those discussion points in my response below.

Point 1: Please provide important information. Presently, this section does contain some self-explaining phrases only. Remember that this part is a stand-alone section, allowing future readers to switch to your main text. 

Response 1: P.2

Introduction

Recent changes in the medical paradigm are highlighting the importance of patient-centered communication. However, due to the lack of awareness of medical institutions and of competence in medical personnel, the quality of medical services regarding communication between doctors and patients has not improved. This study analyzes the impact of health communication and medical service quality, service value, and patient satisfaction on revisiting intention for dental clinics. 

Methods

The study subjects were outpatients who were treated at 10 dental clinics in Seoul. The research data were collected using a questionnaire visited the dental clinics from December 1 to December 30, 2016. A total of 600 questionnaires were distributed to 10 dental clinics, 60 copies each, and 570 valid questionnaires were used for analysis. n this study, the structural influence of factors was determined using structural equation modeling.

Results

The factors influencing service value were reliability (0.364), expertise (0.319), and communication by a doctor (0.224). In addition, the factors influencing patient satisfaction were in the order of reliability (0.258), tangibility (0.192), and communication by a doctor (0.163). On the other hand, service value had a positive effect on patient satisfaction, and patient satisfaction was found to influence dental clinics the reuse intention.

Conclusion

Providing accurate medical services to inpatients based on smooth communication between doctors and patients will have a positive effect on improving patient satisfaction. In addition, the doctor will be able to attract long-term customers by increasing the patients to revisiting through patient-oriented communication.

Point 2: Remember to elaborate both aims and objectives more clearly. There must be a deducable null hypothesis, reasonable and indisputable.

Response 2: P.4

Recently, the concept of service quality, patient satisfaction, and the relationship between reuse intention and service value has been added and considered. Existing prior studies mainly consisted of the relationship between service quality and satisfaction (service quality → satisfaction), service quality and reuse (service quality → reuse intention), service quality and satisfaction, and reuse intention (service quality → satisfaction → reuse intention). However, most of them deal with medical service quality, patient satisfaction, and reuse intention individually, and few studies analyze the relationship between them [6-8].

The purpose of this study is to analyze the structural relationship between the satisfaction of the quality of medical service of outpatients visiting the dental clinic and the reuse intention.

Point 3: Re your research model, this must be understandable for every reader. Just depicting some kind of chart would not seem appropriate. Please remember to guide the reader. 

Response 3: P.5

The purpose of this study was to analyze the structural relationship between health communication and medical service quality, service value, patient satisfaction, and medical institution reuse intention. Therefore, health communication was divided into communication by doctor and communication by assistant, and medical service quality was classified into expertise, reliability, tangibility, and accessibility. In addition, the purpose of this study is to analyze whether health communication and medical service quality influence patient's reuse intention through mediating patient satisfaction and service value. The research model of this study is as follows (Fig 1).

Point 4: Collecting data from December 1 to December 30, 2016 would seem outdated, wouldn't it? 

Response 4: P.5-6

Since research on measuring the quality of health communication and medical services for dental clinics has been insignificant, it is expected that this data will be able to derive a plan to increase dental clinic revisiting intention.

Point 5: This section must provide methodology. Do not give a literature review here. 

Response 5:

It's corrected. 

Point 6: As always, please provide general names with your text, followed by (brand names; manufacturer, city, country) in parentheses. Please provide full information with ALL materials and methodologies. 

Response 6: P.7

SPSS 25.0 (SPSS Inc., Chicago, IL, USA) and Amos 18.0 (SPSS Inc., Chicago, IL, USA) software

Point 7: To test internal consistency among the measurement items, a reliability verification was conducted using the Cronbach's α value. To measure construct validity, a factor analysis was performed using the Varimax mode." This obviously refers to methodology. Please revise. 

Response 7: P.7

Second, to test internal consistency among the measurement items, a reliability verification was conducted using the Cronbach's α value, and to measure construct validity, factor analysis was performed using the Varimax mode.

Point 8: Table 3 shows the results of the CFA for the model used in this study." Again, this would not seem sufficient. Do you expect the reader to analyze these data?

Response 8: P.10

To evaluate the appropriateness of CFA, χ₂ value, the p-value for χ₂ value, TLI(Tucker Lewis Index), CFI(Comparative Fit Index), and RMSEA(Root Mean Square Error of Approximation) were used. The model goodness of fit for the measurement model was that the coefficient values estimated as χ₂=1712.643 (df=783, P=0.001), TLI=0.918, CFI=0.926 were 0.9 or higher, and overall, the model fits were excellent. In addition, RMSEA=0.063 is less than 0.08, making the factor analysis reasonable.

Point 9: This would not seem well-elaborated. Please discuss outcome, provide insight thoughts on the methodology, and speculate on future research directions. As with the other sections, this part must be re-prganized. 

Response 9: P.15-16

These results are the same as those of the previous studies that the reliability of the treatment results improves patient satisfaction and service value. In addition, it was found that communication through the empathy of hospital staff was consistent with the results of previous studies that had a great impact on patient satisfaction and reuse intention [7, 35, 36].

At this point, the limitations of this study and future research directions should be noted. First, questionnaires were distributed and collected from outpatients only in certain dental clinics located in Seoul. Because the distribution status of dental institutions differs from region to region, the results of this study cannot easily be generalized to all dental institutions. If the analysis were to be repeated including dental clinics located in the provinces, accessibility would also affect patient satisfaction and service value. Accordingly, any follow-up studies should attempt to overcome regional limitations. 

Second, since there are many non-payment items in dental treatment, it is necessary to examine what effect these items have on the reuse intention for medical services by considering this specificity of dental treatment and including price factors like medical expenses. Particularly, as implant procedures have been increasing recently, an analysis could be conducted by dividing subjects into two categories: beneficiaries and non-beneficiaries of national health insurance. This method could then consider and analyze the impact of medical expenses on reuse intention. Based on this, the effect of transparent disclosure concerning medical expenses through communication between doctors and patients and its effect on patient satisfaction can be understood. 

Third, this study focused on general dental clinics without classifying them into clinics and network hospitals, and the structural model was applied on communication, quality of medical service, patient satisfaction, service value, and reuse intention. However, given that the number of network hospitals have recently been rapidly increasing and the co-branded opening in the network type targeting specific patients has become generalized, research should be conducted on image factors, which may have a direct effect on patient satisfaction and service value. 

Point 10: Do not simply repeat the results here. This section must provide a reasonable extension of your outcome. Strictly stick to your aims here. 

Response 10: P.17

Therefore, it is considered that the following measures are necessary to increase the satisfaction of patients who have visiting dental clinics and to increase the reuse intention. First, it is necessary to increase the patient's reliability in the doctor by providing accurate information about the process to patients. The reliability of the patient's doctor affects the outpatients and affects patient satisfaction and reuse intention. Therefore, it is necessary for doctors and patients to exchange information on treatment plans to improve reliability and to increase patient satisfaction through smooth communication about the treatment process. Second, patient satisfaction and service value should be increased through active communication by doctors. In the case of dental clinics, patient-centered communication is important because there are many mild diseases and many outpatients. In other words, when choosing a dental clinic, it appears that more consideration is given to the kindness of doctors and nurses. Therefore, it is necessary to strengthen communication with patients and increase service value through an in-depth explanation of diseases. Third, reuse intention should be increased through patient-centered communication. The patient is important to recognize not only the quality of the medical service but also the process in which the medical service is delivered. Therefore, not only the professional medical service provided by the medical staff but also the interaction or communication with the medical staff while the medical service is being provided can be more satisfied and the service value can be increased. Therefore, doctors will be able to enhance communication and increase service value to increase patient satisfaction. This will ultimately increase the patient's reuse intention and attract long-term customers.

Point 11: Please revise for uniform formatting. Stick to the Guidelines. 

Response 11:

It's corrected. 

I hope that the considerable changes made to my research paper coupled with our above arguments will convince the reviewers in reconsidering our manuscript for publication in your journal. I thank you in advance for your kind and thorough attention in the review of my work.

Best regards,

Response to Reviewer Comments

I wish to thank the reviewers for their constructive feedback. The reviewers point out some remaining elements requiring modifications or clarifications in order to validate my manuscript for publication. My manuscript was thoroughly reviewed and updated according to these pertinent remarks. I would like to illustrate these modifications and address those discussion points in my response below.

Point 1: The paper could be enhanced by describing existing knowledge on this topic in the background section and making it more relevant to dentistry. Also there are inconsistencies in calling medical services versus dental clinics is confusing. It is better to stick with dental services and dental clinics if the study was conducted in these settings and to reduce ambiguity. 

Response 1: P.3

A dental clinic is a subject where patients can receive insurance benefits regardless of the referral form, so lighter patients tend to use higher-level hospitals more easily than other subjects. It is necessary to attract them through patient-centered communication.

Point 2: Also, please describe briefly the types of dental care provided in the participating clinics. 

Response 2: P.4

Dental medical services are mainly provided of dental caries, implants, orthodontics, and oral care, and since they require continuous management, they tend to continue to receive treatment in clinics treated initially. Therefore, in the case of dental clinics, it is important to identify factors affecting the patient's reuse intention.

Point 3: Finally, the current description the background has a lot of redundant material, which can be shortened. 

Response 3:

It's corrected.

Point 4: The methods section again have way too much details and could be difficult for the reader to have a good grasp of the approach. Also, while the figure displaying the research model gives the impression of health communication as an independent variable, it appears health communication is a dependent variable. 

Response 4: P.5

The purpose of this study was to analyze the structural relationship between health communication and medical service quality, service value, patient satisfaction, and medical institution reuse intention. Therefore, health communication was divided into communication by doctor and communication by assistant, and medical service quality was classified into expertise, reliability, tangibility, and accessibility. In 

 addition, the purpose of this study is to analyze whether health communication and medical service quality influence patient's reuse intention through mediating patient satisfaction and service value. The research model of this study is as follows (Fig 1).

Point 5: The results section could also be improved by organizing results related to the research question and assessing the reliability of the survey questions separately. Also, clarity can be improved by being concise. 

Response 5: P.10-12

To evaluate the appropriateness of CFA, χ₂ value, the p-value for χ₂ value, TLI(Tucker Lewis Index), CFI(Comparative Fit Index), and RMSEA(Root Mean Square Error of Approximation) were used. The model goodness of fit for the measurement model was that the coefficient values estimated as χ₂=1712.643 (df=783, P=0.001), TLI=0.918, CFI=0.926 were 0.9 or higher, and overall, the model fits were excellent. In addition, RMSEA=0.063 is less than 0.08, making the factor analysis reasonable.

Table 3. Confirmatory factor analysis.

Factor Path Estimate S.E T p-value Standardized Estimate SMC

Health communication Doctor1 ← Health communication 1.000 0.773 0.599

 Doctor2 ← Health communication 1.183 0.074 16.024 *** 0.852 0.725

 Doctor3 ← Health communication 1.009 0.067 15.145 *** 0.814 0.660

 Doctor4 ← Health communication 1.099 0.072 15.298 *** 0.821 0.674

 Doctor5 ← Health communication 1.129 0.071 15.811 *** 0.843 0.714

 Doctor6 ← Health communication 0.872 0.081 10.783 *** 0.372 0.610

 Doctor7 ← Health communication 0.886 0.082 10.772 *** 0.371 0.609

 Assistant1 ← Health communication 1.000 0.832 0.754

 Assistant2 ← Health communication 0.980 0.054 18.112 *** 0.850 0.743

 Assistant3 ← Health communication 0.999 0.053 18.931 *** 0.874 0.763

 Assistant4 ← Health communication 1.049 0.057 18.532 *** 0.862 0.722

 Assistant5 ← Health communication 1.044 0.056 18.743 *** 0.868 0.692

Expertise Expertise1 ← Expertise 1.000 0.839 0.703

 Expertise2 ← Expertise 0.972 0.054 18.004 *** 0.857 0.734

 Expertise3 ← Expertise 1.138 0.055 18.952 *** 0.889 0.791

Reliability Reliability1 ← Reliability 1.000 0.896 0.802

 Reliability2 ← Reliability 1.059 0.043 24.686 *** 0.918 0.843

 Reliability3 ← Reliability 1.024 0.051 20.198 *** 0.838 0.702

 Reliability4 ← Reliability 1.046 0.051 20.312 *** 0.840 0.706

Tangibility Tangibility1 ← Tangibility 1.000 0.754 0.568

 Tangibility2 ← Tangibility 1.153 0.085 13.611 *** 0.765 0.585

 Tangibility3 ← Tangibility 1.227 0.077 15.998 *** 0.882 0.778

 Tangibility4 ← Tangibility 1.239 0.078 15.955 *** 0.880 0.774

 Tangibility5 ← Tangibility 1.306 0.086 15.130 *** 0.839 0.705

 Tangibility6 ← Tangibility 0.979 0.110 8.924 *** 0.271 0.520

Accessibility Accessibility1 ← Accessibility 1.000 0.902 0.813

 Accessibility2 ← Accessibility 1.083 0.053 20.472 *** 0.934 0.872

 Accessibility3 ← Accessibility 0.874 0.067 13.048 *** 0.656 0.430

 Accessibility4 ← Accessibility 0.605 0.081 7.493 *** 0.271 0.520

Service value Service1 ← Service value 1.000 0.941 0.885

 Service2 ← Service value 1.014 0.034 30.273 *** 0.927 0.859

 Service3 ← Service value 0.907 0.039 22.974 *** 0.845 0.715

 Service4 ← Service value 0.801 0.045 17.679 *** 0.751 0.564

 Service5 ← Service value 0.932 0.039 23.868 *** 0.858 0.736

Patient

satisfaction Satisfaction1 ← Satisfaction 1.000 0.908 0.825

 Satisfaction2 ← Satisfaction 1.025 0.035 28.949 *** 0.942 0.888

 Satisfaction3 ← Satisfaction 1.080 0.036 29.909 *** 0.952 0.907

 Satisfaction4 ← Satisfaction 1.057 0.042 25.243 *** 0.899 0.809

Reuse intention Reuse1 ← Reuse intention 1.000 0.944 0.892

 Reuse2 ← Reuse intention 1.023 0.026 39.650 *** 0.972 0.945

 Reuse3 ← Reuse intention 0.968 0.028 34.935 *** 0.947 0.897

 Reuse4 ← Reuse intention 0.978 0.033 29.875 *** 0.912 0.832

Sample Size, n=570, χ₂=1197.727, df=532, χ₂/df=2.25, TLI=.931, CFI=.938, RMSEA=.065

χ₂=Chi-square statistic, DF=Degrees of freedom, TLI=Tucker Lewis index, CFI= Comparative fit index, RMSEA=Root mean square error of approximation.

Point 6: The discussion section again can be enhanced by describing major findings concisely and clearly. It is also not clear what types of medical services dentists provide. Also, references need to be cited appropriately.

Response 6: P.14-15

It's corrected. (Reference added).

Particularly, dental treatments such as dental caries, implants, and orthodontics are usually provided treatments for 2 to 3 years—rather than one-off treatments—they have a strong tendency to maintain services.

I hope that the considerable changes made to my research paper coupled with our above arguments will convince the reviewers in reconsidering our manuscript for publication in your journal. I thank you in advance for your kind and thorough attention in the review of my work.

Best regards, 

Response to Reviewer Comments

I wish to thank the reviewers for their constructive feedback. The reviewers point out some remaining elements requiring modifications or clarifications in order to validate my manuscript for publication. My manuscript was thoroughly reviewed and updated according to these pertinent remarks. I would like to illustrate these modifications and address those discussion points in my response below.

Point 1: The discussion section: lacks referencing as the authors stated several times (previous studies) without giving reference to which study? 

Response 1: 

It's corrected. (Reference added)

Point 2: For the conclusion section: the limitations should be stated within the discussion section and the conclusion should summarise only the key result and future studies if required. 

Response 2: P.17

Therefore, it is considered that the following measures are necessary to increase the satisfaction of patients who have visiting dental clinics and to increase the reuse intention. First, it is necessary to increase the patient's reliability in the doctor by providing accurate information about the process to patients. The reliability of the patient's doctor affects the outpatients and affects patient satisfaction and reuse intention. Therefore, it is necessary for doctors and patients to exchange information on treatment plans to improve reliability and to increase patient satisfaction through smooth communication about the treatment process. Second, patient satisfaction and service value should be increased through active communication by doctors. In the case of dental clinics, patient-centered communication is important because there are many mild diseases and many outpatients. In other words, when choosing a dental clinic, it appears that more consideration is given to the kindness of doctors and nurses. Therefore, it is necessary to strengthen communication with patients and increase service value through an in-depth explanation of diseases. Third, reuse intention should be increased through patient-centered communication. The patient is important to recognize not only the quality of the medical service but also the process in which the medical service is delivered. Therefore, not only the professional medical service provided by the medical staff but also the interaction or communication with the medical staff while the medical service is being provided can be more satisfied and the service value can be increased. Therefore, doctors will be able to enhance communication and increase service value to increase patient satisfaction. This will ultimately increase the patient's reuse intention and attract long-term customers.

I hope that the considerable changes made to my research paper coupled with our above arguments will convince the reviewers in reconsidering our manuscript for publication in your journal. I thank you in advance for your kind and thorough attention in the review of my work.

Best regards,

---

## [Decision Letter · Decision Letter 1]

3 Sep 2020

PONE-D-19-29416R1

Factors affecting revisiting intention for medical services at dental clinics

PLOS ONE

Dear Dr. lee,

Thank you for submitting your manuscript to PLOS ONE. After careful consideration, we feel that it has merit but still does not fully meet PLOS ONE’s publication criteria as it currently stands. Therefore, we invite you to submit a revised version of the manuscript that addresses the points raised during the review process.

Having intensively reviewed your draft, your revised and re-submitted draft still would not seem satisfying. I have double checked your submitted draft, and, in particular, you should follow the R #1 comments, to finalize your paper convincingly, and to meet both PLOS ONE's quality standards and our readership's expectations. Please note that a further non-convincing revision (not considered acceptable with regard to language, reviewers' constructive criticism, content, generalizable outcome, and/or Authors' Guidelines) will lead to outright reject.

We look forward to receiving your revised manuscript.

Kind regards,

Andrej M Kielbassa

Academic Editor

PLOS ONE

Reviewers' comments:

Reviewer's Responses to Questions

**Comments to the Author**

1. If the authors have adequately addressed your comments raised in a previous round of review and you feel that this manuscript is now acceptable for publication, you may indicate that here to bypass the “Comments to the Author” section, enter your conflict of interest statement in the “Confidential to Editor” section, and submit your "Accept" recommendation.

Reviewer #1: (No Response)

Reviewer #3: All comments have been addressed

2. Is the manuscript technically sound, and do the data support the conclusions?

Reviewer #1: No

Reviewer #3: Yes

3. Has the statistical analysis been performed appropriately and rigorously? 

Reviewer #1: Yes

Reviewer #3: Yes

4. Have the authors made all data underlying the findings in their manuscript fully available?

Reviewer #1: Yes

Reviewer #3: Yes

5. Is the manuscript presented in an intelligible fashion and written in standard English?

Reviewer #1: No

Reviewer #3: Yes

6. Review Comments to the Author

Reviewer #1: General remark

- This submission has improved to some extent, no doubt. However, the authors have failed to follow all the reviewers' recommendations, and this would seem astonishing. Please note that a reviewer invests much time in your submission, to improve understanding and perception. Ignoring those recommendations generally is not appreciated.

Abstract

- Please add EXACT results, and provide P values.

Intro

- Please note that each statement of facts must be accompanied by a reference. See, for example, "The importance of communication in medical services has been highlighted before". Revise thoroughly.

- "Therefore, this study analyzes (...)." You have done this already, right? Please switch to past tense. Revise thoroughly throughout your draft.

- Again, , as recommended previously, please provide a null hypothesis. Remember that H0 must be reasonable and deducible from the foregoing thoughts.

Meths

- "The research model of this study is as follows:" Again, as recommended previously, you should guide the readers. This means that each figure must be accompanied by text, thus explaining the authors' intention.

- "However, there were many incomplete questionnaires." What does this mean? How many is "many"?

- Again, please note that this section is not supposed to provide a literature review. This ha been recommended previously, and the authors obviously do not want to follow this aspect. See "In 1985, a study by Parasuraman et al. on the quality of medical services provided five categories of quality. These quality categories include (...)."

- Same with "In 1992, Cronin and Taylor attempted to measure service quality based on (...)." Please revise thoroughly, and do not mix section contents.

- Please provide manufacturers of the software used.

Results

- Please double check legends of both tables and figures, and revise for readability.

- Instead of indicating ***, please provide exact P values.

Disc

- "This differs from the findings of previous studies that state (...)." What studies do you refer to here? Again, please note that each such statement must be accompanied by a reference.

- Same with "Furthermore, the results also differed from previous study results stating that (...)." Please note that a reviewer's task is not considered to co-author your manuscript.

- Same with "This is consistent with the findings of previous studies (...)." Again, this revised and re-submitted draft would not seem convincingly elaborated.

Concl

- Do not repeat methodology or results here. Instead, provide a reasonable extension of your outcome which must stick to the aims of your study.

- Note that aspects like "limitations" must be given with the Discussion section.

Refs

- Still the references would not seem to follow the Journal's Guidelines for Authors. No doubt, it would seem hard to understand why the authors do not want to follow those guidelines.

In total, this revised version would not seem ready to proceed.

-

Reviewer #3: Thank you for addressing the comments fully, which made the manuscript sound technically and scientifically.

7. PLOS authors have the option to publish the peer review history of their article (what does this mean?). If published, this will include your full peer review and any attached files.

Reviewer #1: No

Reviewer #3: No

---

## [Author Response · Author response to Decision Letter 1]

19 Oct 2020

Response to Reviewer Comments

I wish to thank the reviewers for their constructive feedback. The reviewers point out some remaining elements requiring modifications or clarifications in order to validate my manuscript for publication. My manuscript was thoroughly reviewed and updated according to these pertinent remarks. I would like to illustrate these modifications and address those discussion points in my response below.

Point 1: Abstract - Please add EXACT results, and provide P values.

Response 1: P.2

The factors influencing service value were reliability (β=0.364, p=0.000), expertise (β=0.319, p=0.000), communication by a doctor (β=0.224, p=0.000) and tangibles (β=0.136, p=0.032). In addition, the factors influencing patient satisfaction were in the order of reliability (β=0.258, p=0.000), tangibility (β=0.192, p=0.000), communication by a doctor (β=0.163, p=0.001) and Expertise (β=0.122, p=0.075). On the other hand, service value (β=0.438, p=0.000) had a positive effect on patient satisfaction, and patient satisfaction (β=0.383, p=0.000) was found to influence dental clinics the reuse intention.

Point 2: Intro - Please note that each statement of facts must be accompanied by a reference. See, for example, "The importance of communication in medical services has been highlighted before". Revise thoroughly.

Response 2: 

This part was removed under reviewer advice from a previous revision. Thank you for reviewing my paper so that it can be published.

Point 3: Intro - "Therefore, this study analyzes (...)." You have done this already, right? Please switch to past tense. Revise thoroughly throughout your draft.

Response 3: P.4

This study analyzes the effect of health communication and service quality on service value, patient satisfaction, and reuse intention, focusing on dental clinics with high patient interaction.

Point 4: Intro - Again, , as recommended previously, please provide a null hypothesis. Remember that H0 must be reasonable and deducible from the foregoing thoughts.

Response 4: P.4

To this end, in this study, the null hypothesis is that communication by doctors and assistant staff affects the reuse intention medical institutions with service value and patient satisfaction as parameters. In addition, this study aims to analyze by setting the null hypothesis that the medical services quality affects the reuse intention medical institutions by mediating service value and patient satisfaction.

Point 5: Meths - "The research model of this study is as follows:" Again, as recommended previously, you should guide the readers. This means that each figure must be accompanied by text, thus explaining the authors' intention.

Response 5: P.5

The research model of this study is as follows (Fig 1). This study attempted to analyze the structural impact relationship between health communication and the intention to reuse medical institutions through patient satisfaction and service value. For this, health communication and medical service quality were used as independent variables, and the reuse intention was used as the dependent variable. In addition, patient satisfaction and service value were used as parameters. Communication by doctors and by assistants was selected as sub-items of health communication, which is an independent variable, and expertise, reliability, tangibility, and accessibility were selected and measured as sub-items of medical service quality.

Point 6: Meths - "However, there were many incomplete questionnaires." What does this mean? How many is "many"?

Response 6: P.6

However, there were 30 incomplete questionnaires

Point 7: Meths - Again, please note that this section is not supposed to provide a literature review. This ha been recommended previously, and the authors obviously do not want to follow this aspect. See "In 1985, a study by Parasuraman et al. on the quality of medical services provided five categories of quality. These quality categories include (...)."

Response 7: 

This part was removed under reviewer advice from a previous revision. Thank you for reviewing my paper so that it can be published.

Point 8: Meths - Same with "In 1992, Cronin and Taylor attempted to measure service quality based on (...)." Please revise thoroughly, and do not mix section contents.

Response 8: 

This part was removed under reviewer advice from a previous revision. Thank you for reviewing my paper so that it can be published.

Point 9: Meths - Please provide manufacturers of the software used.

Response 9: P.8

The data analysis was conducted using SPSS 25.0 (SPSS Inc., IBM, Chicago, IL, USA) and Amos 18.0 (SPSS Inc., IBM, Chicago, IL, USA) software.

Point 10: Results - Please double check legends of both tables and figures, and revise for readability.

Response 10: P.13-14

Table 3. 

***p<0.001, S.E=Standard error, T=t-value, β=Standardized coefficients, SMC=Squared Multiple Correlation.

Table4. 

***p<0.001, X2=Chi-square statistic, DF=Degrees of freedom, TLI=Tucker Lewis index, CFI= Comparative fit index, RMSEA=Root mean square error of approximation.

Table5.

*p<0.1, **p<0.05, ***p<0.001, B=Unstandardized coefficients, β=Standardized coefficients, S.E=Standard error, T=t-value

  

Point 11: Results - Instead of indicating ***, please provide exact P values.

Response 11: P.13

Factor Path Estimate S.E T p-value β SMC

Health communication Doctor1 ← Health communication 1.000 0.773 0.599

 Doctor2 ← Health communication 1.183 0.074 16.024 0.001 0.852 0.725

 Doctor3 ← Health communication 1.009 0.067 15.145 0.001 0.814 0.660

 Doctor4 ← Health communication 1.099 0.072 15.298 0.001 0.821 0.674

 Doctor5 ← Health communication 1.129 0.071 15.811 0.001 0.843 0.714

 Doctor6 ← Health communication 0.872 0.081 10.783 0.001 0.372 0.610

 Doctor7 ← Health communication 0.886 0.082 10.772 0.001 0.371 0.609

 Assistant1 ← Health communication 1.000 0.832 0.754

 Assistant2 ← Health communication 0.980 0.054 18.112 0.001 0.850 0.743

 Assistant3 ← Health communication 0.999 0.053 18.931 0.001 0.874 0.763

 Assistant4 ← Health communication 1.049 0.057 18.532 0.001 0.862 0.722

 Assistant5 ← Health communication 1.044 0.056 18.743 0.001 0.868 0.692

Expertise Expertise1 ← Expertise 1.000 0.839 0.703

 Expertise2 ← Expertise 0.972 0.054 18.004 0.001 0.857 0.734

 Expertise3 ← Expertise 1.138 0.055 18.952 0.001 0.889 0.791

Reliability Reliability1 ← Reliability 1.000 0.896 0.802

 Reliability2 ← Reliability 1.059 0.043 24.686 0.001 0.918 0.843

 Reliability3 ← Reliability 1.024 0.051 20.198 0.001 0.838 0.702

 Reliability4 ← Reliability 1.046 0.051 20.312 0.001 0.840 0.706

Tangibility Tangibility1 ← Tangibility 1.000 0.754 0.568

 Tangibility2 ← Tangibility 1.153 0.085 13.611 0.001 0.765 0.585

 Tangibility3 ← Tangibility 1.227 0.077 15.998 0.001 0.882 0.778

 Tangibility4 ← Tangibility 1.239 0.078 15.955 0.001 0.880 0.774

 Tangibility5 ← Tangibility 1.306 0.086 15.130 0.001 0.839 0.705

 Tangibility6 ← Tangibility 0.979 0.110 8.924 0.001 0.271 0.520

Accessibility Accessibility1 ← Accessibility 1.000 0.902 0.813

 Accessibility2 ← Accessibility 1.083 0.053 20.472 0.001 0.934 0.872

 Accessibility3 ← Accessibility 0.874 0.067 13.048 0.001 0.656 0.430

 Accessibility4 ← Accessibility 0.605 0.081 7.493 0.001 0.271 0.520

Service value Service1 ← Service value 1.000 0.941 0.885

 Service2 ← Service value 1.014 0.034 30.273 0.001 0.927 0.859

 Service3 ← Service value 0.907 0.039 22.974 0.001 0.845 0.715

 Service4 ← Service value 0.801 0.045 17.679 0.001 0.751 0.564

 Service5 ← Service value 0.932 0.039 23.868 0.001 0.858 0.736

Patient

satisfaction Satisfaction1 ← Satisfaction 1.000 0.908 0.825

 Satisfaction2 ← Satisfaction 1.025 0.035 28.949 0.001 0.942 0.888

 Satisfaction3 ← Satisfaction 1.080 0.036 29.909 0.001 0.952 0.907

 Satisfaction4 ← Satisfaction 1.057 0.042 25.243 0.001 0.899 0.809

Reuse intention Reuse1 ← Reuse intention 1.000 0.944 0.892

 Reuse2 ← Reuse intention 1.023 0.026 39.650 0.001 0.972 0.945

 Reuse3 ← Reuse intention 0.968 0.028 34.935 0.001 0.947 0.897

 Reuse4 ← Reuse intention 0.978 0.033 29.875 0.001 0.912 0.832

***p<0.001, S.E=Standard error, T=t-value, β=Standardized coefficients, SMC=Squared Multiple Correlation. 

  

Point 12: Disc - "This differs from the findings of previous studies that state (...)." What studies do you refer to here? Again, please note that each such statement must be accompanied by a reference. - Same with "Furthermore, the results also differed from previous study results stating that (...)." Please note that a reviewer's task is not considered to co-author your manuscript.

Response 12: P.16

These results are shown in Chang et al. (2013), the doctor's communication attitude affects the overall medical satisfaction of patients, and the medical service quality and the patient's reliability are similar to the results of a study that affects patient satisfaction. Also, Rashid et al. The health communication of doctors in (2014) were similar to the results of a study that showed greater satisfaction to patients than clinical competence [31, 32].

This is, Ehsan et al. (2015) The interaction between the medical staff and the patient affects the overall satisfaction of medical services, and it is different from the research results that the more the communication by doctors and assistant staff with the patients more smoothly, the patient satisfaction is improved [35]. In addition, Fellani Danasra et al. In (2011), most of the patients receiving dental treatment want to talk to the assistant staff about their discomfort, which is different from the research results that have an effect on the patient's reuse intention of dental institutions [36].

Point 13: Disc - Same with "This is consistent with the findings of previous studies (...)." Again, this revised and re-submitted draft would not seem convincingly elaborated.

Response 13: P.16-17

In Seema (2011), patient satisfaction is similar to the results of a study that observes the treatment process and maintains the continuity of treatment, and that patient satisfaction closely affects the reuse intention a medical institution [37]. Also, Anang et al. In (2019), research results show that the quality of service at a medical institution affects patient satisfaction [38]. Quality of service has a significant effect on the reuse intention and is similar to previous research results that show a significant correlation between satisfaction of outpatients, quality of service, and reuse intention [38-40].

Point 14: Concl - Do not repeat methodology or results here. Instead, provide a reasonable extension of your outcome which must stick to the aims of your study.

Response 14: 

The revision was made according to the reviewer's opinion in the previous revision. Thank you for reviewing my paper so that it can be published.

Point 15: Concl - Note that aspects like "limitations" must be given with the Discussion section.

Response 15: 

The revision was made according to the reviewer's opinion in the previous revision. Thank you for reviewing my paper so that it can be published.

Point 16: Refs - Still the references would not seem to follow the Journal's Guidelines for Authors. No doubt, it would seem hard to understand why the authors do not want to follow those guidelines.

Response 16: P.19

Modified according to reviewer's advice. Thank you for reviewing my paper so that it can be published.

I hope that the considerable changes made to my research paper coupled with our above arguments will convince the reviewers in reconsidering our manuscript for publication in your journal. I thank you in advance for your kind and thorough attention in the review of my work.

Best regards,

Response to Reviewer Comments

I wish to thank the reviewers for their constructive feedback. The reviewers point out some remaining elements requiring modifications or clarifications in order to validate my manuscript for publication. My manuscript was thoroughly reviewed and updated according to these pertinent remarks. I would like to illustrate these modifications and address those discussion points in my response below.

Point 1: Thank you for addressing the comments fully, which made the manuscript sound technically and scientifically.

Response 1: 

Thank you for reviewing my paper so that it can be published.

I hope that the considerable changes made to my research paper coupled with our above arguments will convince the reviewers in reconsidering our manuscript for publication in your journal. I thank you in advance for your kind and thorough attention in the review of my work.

Best regards,

---

## [Decision Letter · Decision Letter 2]

2 Dec 2020

PONE-D-19-29416R2

Factors affecting revisiting intention for medical services at dental clinics

PLOS ONE

Dear Dr. Lee,

Thank you for submitting your manuscript to PLOS ONE. After careful consideration, we feel that it has merit but does not fully meet PLOS ONE’s publication criteria as it currently stands. Therefore, we invite you to submit a revised version of the manuscript that addresses the points raised during the review process.

Having intensively reviewed your draft, our reviewer has indicated that your submitted draft would not seem satisfying. I have double checked your submitted draft, to come to a more balanced decision. Indeed, you should follow the reviewer's comments, to finalize your paper convincingly, and to meet both PLOS ONE's quality standards and our readership's expectations. Please note that a non-convincing revision (not considered acceptable with regard to language, reviewers' constructive criticism, content, generalizable outcome, and/or Authors' Guidelines) will lead to outright reject. 

We look forward to receiving your revised manuscript.

Kind regards,

Andrej M Kielbassa

Academic Editor

PLOS ONE

Reviewers' comments:

Reviewer's Responses to Questions

**Comments to the Author**

1. If the authors have adequately addressed your comments raised in a previous round of review and you feel that this manuscript is now acceptable for publication, you may indicate that here to bypass the “Comments to the Author” section, enter your conflict of interest statement in the “Confidential to Editor” section, and submit your "Accept" recommendation.

Reviewer #1: (No Response)

Reviewer #4: (No Response)

2. Is the manuscript technically sound, and do the data support the conclusions?

Reviewer #1: Yes

Reviewer #4: Partly

3. Has the statistical analysis been performed appropriately and rigorously? 

Reviewer #1: Yes

Reviewer #4: No

4. Have the authors made all data underlying the findings in their manuscript fully available?

Reviewer #1: Yes

Reviewer #4: Yes

5. Is the manuscript presented in an intelligible fashion and written in standard English?

Reviewer #1: Yes

Reviewer #4: Yes

6. Review Comments to the Author

Reviewer #1: Abstract

- "n this study, the structural (...)." meaning remains unclear, please revise.

- Please note that "p=0.000" would seem hardly possible. Must read p=0.001 (if this is the exact value), or p<0.001 (if exact result would be p=0.0009, for example), or p<0.0001 (if exact result would be p=0.000006, for example). Please revise thoroughly.

Intro

- "(...) model—where (...)" must read "(...) model — where (...)". Re-edit, and make use of your spacebar.

- Same with "(...) personnel — to a (...)". See also the same formatting shortcoming later on.

- "(...) so lighter patients tend to use higher-level hospitals (...)." Meaning of "lighter" patient remains unclear. Please clarify.

- Do not use unclear symbols with your text, see "(service quality → satisfaction)". Meaning of " → " remains unclear. With your full text, please provide full sentences. In this case, " → " could mean "service quality leads to higher satisfaction". Revise thoroughly.

- Authors have stated a false null hypothesis ("To this end, in this study, the null hypothesis is that communication by doctors and assistant staff affects the reuse intention medical institutions with service value and patient satisfaction as parameters."). Please remember that H0 proposes that there is NO difference between certain characteristics of a population (or data-generating process). Please see definitions on the web, and revise carefully.

- Same with "In addition, this study aims to analyze by setting the null hypothesis that the medical services quality affects the reuse (...)."

- "This study constructed a questionnaire based on the measurement items developed by Bowers et al., Marley et al., and Goleman." Please provide reference numbers after each author name/group.

- "Concerning medical service quality, (...)", and "Patient satisfaction is the cognitive response (...)". With your methods section, do not provide a literature review, and do not provide explanations or definitions. Stick exclusively to your methodology, and re-edit subheadings. Aspects considered necessary for the readership must be provided either with the Intro, or with the Disc section. Revise thoroughly.

- Same with "Service value refers to (...)", and "Reuse intention refers to (...)."

- Do not use legal terms like Inc., and so on. Please delete.

Results

- Again, please revise for minor typos. "(Table1)" must read "(Table 1)", you surely will agree.

- Same with TLI(Tucker Lewis Index), and so on. make use of your spacebar to separate acronyms and full text. Revise thoroughly.

- Again, revise for uniform formatting. Compare "TLI=0.918" and "TLI = 0.910". This clearly is considered the authors task. Always use X = Y, and make use of your spacebar. Revise thoroughly throughout your text.

Disc

- What about H0? Was it rejected or not rejected?

- Again, to facilitate reading, please revise your text for typos. See, for example, "Also, Rashid et al. The health communication (...)". Please note that all (co-)authors must read AND approve your submission before re-submitting your paper. This includes revision of typos. With 4 (!) authors/contributors, the number of minor and major shortcomings should be reducible, don't you agree?

- Same with "This is, Ehsan et al. (2015) The interaction between (...)". This text would seem perfectible.

- Again, same with "In addition, Fellani Danasra et al. In (2011), most of the patients (...)". Revise thoroughly throughout your text, and search some help of a native speaker.

Concl

- Revise to facilitate reading. Do not double terms like "therefore".

Refs

- Authors have failed to uniformly format this section.

- Again, stick to Guidelines for Authors, and consult some recently published papers. See the following example:

Cheng L, Weir MD, Xu HH, Antonucci JM, Lin NJ, Lin-Gibson S, et al. Effect of amorphous calcium phosphate and silver nanocomposites on dental plaque microcosm biofilms. J Biomed Mater Res B Appl Biomater. 2012; 100(5): 1378–1386. https://doi.org/10.1002/jbm.b.32709 PMID: 22566464 Revise thoroughly, and remember that proceeding will not be possible without a complete revision of your draft.

In total, this draft would seem worth following, but clearly is not considered ready to proceed.

Reviewer #4: The manuscript presents extremely relevant data for the organization of health actions; however, some aspects need to be better described.

The title of the manuscript leads us to the understanding of the services offered in dental clinics, however, when reading the text there is little focus specifically on this service, since the text is more centered on doctor and medical services. The suggestion is that the text be described about health services in general, and specifically about dental clinics and the dentist.

The research was carried out on dental clinics and dentists, however, there is little presence of these terms throughout the text, which allows for a confused reading of the text by over-mentioning the terms doctor and medical services.

Abstract

The abstract needs to present the context of dental clinics and not medical services, it is important to highlight the object of the study. It is necessary to review the verbs and present them in the past. The last sentence of the methods is incomplete. Replace p=0.000 with p<0.001, considering that there is no statistical significance equal to zero. In the statistical packages, when checking the output, it is possible to verify the exact significance.

Introduction

The first paragraph has no reference. Review throughout the text to prioritize the use of the terms: dentists and dental clinics. It is necessary to review the null hypothesis presented since this hypothesis generally states that there is no relationship between the studied phenomena. The objectives need to be better described, as there is duplication in the presentation.

Methods

The text does not make it clear what the inclusion criteria were. Were people under 18 included?

The data analysis section needs to be reviewed carefully. It is necessary to describe in detail the analyzes carried out, as well as the criteria used for each type of analysis. What were the criteria and procedures adopted for the factor analysis? What were the post-tests used to assess the adequacy of the model? What criteria are used?

Results

The results related to factor analysis were not presented. What were the communalities, the sample adequacy measures, the variance explained by each factor?

In the instrument used, there are more dimensions than those shown in Table 2. The dimensions “Expertise of assistant staff” and “Responsiveness of the office or clinic” are not listed in the table. The number of items in the Communication by assistant dimension is different on the table and on the instrument.

The instrument contains 48 items; however, it was presented that the analysis was performed with only 38 items, it is necessary to present the reasons that led to the exclusion of 10 items.

In table 5, it is necessary to review the presentation of the p-values (p=0.000). If the p-value was presented, it is not necessary to use symbols to describe the statistical significance.

In figure 2, the expertise dimension has repeated values, it is necessary to correct it. As this is the figure that presents the final model, I suggest that only the relations that were significant for the composition of the final model be presented.

Discussion

In view of the notes made, it is suggested that all dimensions of the final model be addressed in the discussion. It is necessary to review the use of the terms doctor and medical services.

Conclusion

It is highly recommended that the title, objectives and conclusion are related and that the conclusion responds directly to the proposed objectives.

7. PLOS authors have the option to publish the peer review history of their article (what does this mean?). If published, this will include your full peer review and any attached files.

Reviewer #1: No

Reviewer #4: **Yes: **Arthur de Almeida Medeiros

---

## [Author Response · Author response to Decision Letter 2]

16 Jan 2021

Response to Reviewer 1 Comments

I wish to thank the reviewers for their constructive feedback. The reviewers point out some remaining elements requiring modifications or clarifications in order to validate my manuscript for publication. My manuscript was thoroughly reviewed and updated according to these pertinent remarks. I would like to illustrate these modifications and address those discussion points in my response below.

Point 1: "n this study, the structural (...)." meaning remains unclear, please revise.

Response 1: P.2

In this study, the structural influence of factors was determined using structural equation modeling. 

Point 2: Please note that "p=0.000" would seem hardly possible. Must read p=0.001 (if this is the exact value), or p<0.001 (if exact result would be p=0.0009, for example), or p<0.0001 (if exact result would be p=0.000006, for example). Please revise thoroughly.

Response 2: P.2

Results

The factors influencing service value were reliability (β = 0.364, p < 0.001), expertise (β = 0.319, p < 0.001), communication by a doctor (β = 0.224, p < 0.001) and tangibles (β = 0.136, p < 0.05). In addition, the factors influencing patient satisfaction were in the order of reliability (β = 0.258, p < 0.001), tangibility (β = 0.192, p < 0.001), communication by a doctor (β = 0.163, p < 0.001) and Expertise (β = 0.122, p < 0.01). On the other hand, service value (β = 0.438, p < 0.001) had a positive effect on patient satisfaction, and patient satisfaction (β = 0.383, p < 0.001) was found to influence dental clinics the reuse intention.

Point 3: "(...) model—where (...)" must read "(...) model — where (...)". Re-edit, and make use of your spacebar.

Response 3: P.3

Medical services are changing from a disease-centered model to a patient-centered model. In the existing disease-centered model, all the decision-making concerning patient care was conducted based on the expertise of doctors and other medical personnel, but in the patient-centered model, the patient actively participates in their treatment process and their needs and preferences are reflected in care-related decision making [1, 2].

Point 4: Same with "(...) personnel — to a (...)". See also the same formatting shortcoming later on.

Response 4: P.3

Medical services are changing from a disease-centered model to a patient-centered model. In the existing disease-centered model, all the decision-making concerning patient care was conducted based on the expertise of doctors and other medical personnel, but in the patient-centered model, the patient actively participates in their treatment process and their needs and preferences are reflected in care-related decision making [1, 2].

Point 5: "(...) so lighter patients tend to use higher-level hospitals (...)." Meaning of "lighter" patient remains unclear. Please clarify.

Response 5: P.3

In general, in order to receive care in Korea, primary and secondary medical institutions must be visited first, and in case of major ailment patients, a medical referral form is issued and care can be received at tertiary medical institutions. However, most dental clinics are composed of primary and secondary medical institutions, and primary medical institutions can receive treatment at the level of tertiary medical institutions. In addition, most of the dental clinics tend to have outpatients and relatively few patients with severe diseases. Therefore, it is necessary to increase the patient's reuse intention through patient-centered communication because the patient can choose a dental clinics to receive treatment [3].

Point 6: Do not use unclear symbols with your text, see "(service quality → satisfaction)". Meaning of " → " remains unclear. With your full text, please provide full sentences. In this case, " → " could mean "service quality leads to higher satisfaction". Revise thoroughly.

Response 6: P.4

Existing prior studies mainly consisted of the relationship between service quality affects satisfaction, or service quality affects reuse intention. In addition, it is focused on research that service quality affects satisfaction, and satisfaction affects reuse intention.

Point 7:

- Authors have stated a false null hypothesis ("To this end, in this study, the null hypothesis is that communication by doctors and assistant staff affects the reuse intention medical institutions with service value and patient satisfaction as parameters."). Please remember that H0 proposes that there is NO difference between certain characteristics of a population (or data-generating process). Please see definitions on the web, and revise carefully.

- Same with "In addition, this study aims to analyze by setting the null hypothesis that the medical services quality affects the reuse (...)."

Response 7: P.4

To this end, the null hypothesis was established that health communication and medical service quality do not affect the reuse intention of dental clinics by mediating service value and patient satisfaction.

Point 8: "This study constructed a questionnaire based on the measurement items developed by Bowers et al., Marley et al., and Goleman." Please provide reference numbers after each author name/group.

Response 8: P.6

This study constructed a questionnaire based on the measurement items developed by Bowers et al. [12], Marley et al. [13], and Goleman [14].

Point 9: "Concerning medical service quality, (...)", and "Patient satisfaction is the cognitive response (...)". With your methods section, do not provide a literature review, and do not provide explanations or definitions. Stick exclusively to your methodology, and re-edit subheadings. Aspects considered necessary for the readership must be provided either with the Intro, or with the Disc section. Revise thoroughly.

Response 9: P.6 

Concerning medical service quality, the SERVOPERF measurement model of Cronin and Taylor was utilized [17, 18]. To measure service quality using the SERVPERF model, 17 questions were used: three items for expertise, four for reliability, six for tangibility, and four for accessibility [19-21]. 

Patient satisfaction was measured by using the measurement items developed by Westbrook [22], Woodside et al. [23], and Dodd et al. [24].

Point 10: Same with "Service value refers to (...)", and "Reuse intention refers to (...)."

Response 10: P.6

Service value is the physical and emotional value the patient feels through the treatment process and results.

Point 11: Do not use legal terms like Inc., and so on. Please delete.

Response 11: P.8

It's corrected.

IBM, Chicago, IL, USA

Point 12: Again, please revise for minor typos. "(Table1)" must read "(Table 1)", you surely will agree.

Response 12: P.9-10

It's corrected.

Point 13: Same with TLI(Tucker Lewis Index), and so on. make use of your spacebar to separate acronyms and full text. Revise thoroughly.

Response 13: P.11

It's corrected.

Tucker Lewis Index (TLI), Comparative Fit Index (CFI), and Root Mean Square Error of Approximation (RMSEA)

Point 14: Again, revise for uniform formatting. Compare "TLI=0.918" and "TLI = 0.910". This clearly is considered the authors task. Always use X = Y, and make use of your spacebar. Revise thoroughly throughout your text.

Response 14: P.11, P.14

The model goodness of fit for the measurement model was that the coefficient values estimated as χ₂ = 1712.643 (df = 783, P < 0.001), TLI = 0.918, CFI = 0.926 were 0.9 or higher, and overall, the model fits were excellent. In addition, RMSEA = 0.063 is less than 0.08, making the factor analysis reasonable.

The result of analyzing the model used in the study showed that χ₂ = 1653.662, TLI = 0.910, CFI = 0.917, and RMSEA = 0.072, indicating that the values of the indexes are generally excellent and satisfactory. The goodness-of-fit of the research model is shown in Table 4.

Point 15: What about H0? Was it rejected or not rejected?

Response 15: P.16

As a result of the study, health communication and quality of medical service influenced the reuse intention of dental clinics by mediating patient satisfaction and service value, thus rejecting the null hypothesis and adopting the alternative hypothesis. The detailed study results are as follows.

Point 16: 

- Again, to facilitate reading, please revise your text for typos. See, for example, "Also, Rashid et al. The health communication (...)". Please note that all (co-)authors must read AND approve your submission before re-submitting your paper. This includes revision of typos. With 4 (!) authors/contributors, the number of minor and major shortcomings should be reducible, don't you agree?

- Same with "This is, Ehsan et al. (2015) The interaction between (...)". This text would seem perfectible.

"This is, Ehsan et al. (2015) The interaction between (...)"

- Again, same with "In addition, Fellani Danasra et al. In (2011), most of the patients (...)". Revise thoroughly throughout your text, and search some help of a native speaker.

Response 16: P.16

Also, Rashid et al. (2014), the health communication of doctors in were similar to the results of a study that showed greater satisfaction to patients than clinical competence [34].

This is, Ehsan et al. (2015), showed that the smoother communication between doctors and the assistant staff was found to be different from the research results that are linked to patient satisfaction [37]. In addition, Fellani Danasra et al. (2011), most of the patients receiving dental treatment want to talk to the assistant staff about their discomfort in treatment, which is different from the research results that it affects the patient's reuse intention of dental clinics [38].

Point 17: Revise to facilitate reading. Do not double terms like "therefore".

Response 17: P.18

This study aimed to analyze the influencing factors of the reuse of medical services, employing data from patients visiting dental clinics located in Seoul. The results showed that reliability and communication by doctors affected service value and patient satisfaction, which had an effect on reuse intention. It is considered that the following measures are necessary to increase the satisfaction of patients who have visiting dental clinics and to increase the reuse intention. Dental clinics should provide appropriate medical services to outpatients, which is based on smooth communication between doctors and patients. Additionally, doctors having an attitude of respect toward the patient may affect patient satisfaction. Doctors providing medical treatment information to patients with a friendly and respectful attitude rather than an authoritarian attitude may be an effective strategy for dental medical institutions to attract long-term customers.

Point 18: Refs

- Authors have failed to uniformly format this section.

- Again, stick to Guidelines for Authors, and consult some recently published papers. See the following example:

Cheng L, Weir MD, Xu HH, Antonucci JM, Lin NJ, Lin-Gibson S, et al. Effect of amorphous calcium phosphate and silver nanocomposites on dental plaque microcosm biofilms. J Biomed Mater Res B Appl Biomater. 2012; 100(5): 1378–1386. https://doi.org/10.1002/jbm.b.32709 PMID: 22566464 Revise thoroughly, and remember that proceeding will not be possible without a complete revision of your draft.

Response 18:

It's corrected.

Point 19: In total, this draft would seem worth following, but clearly is not considered ready to proceed.

Response 19: 

Thank you for reviewing my paper so that it can be published.

I hope that the considerable changes made to my research paper coupled with our above arguments will convince the reviewers in reconsidering our manuscript for publication in your journal. I thank you in advance for your kind and thorough attention in the review of my work.

Best regards,

Response to Reviewer 2 Comments

I wish to thank the reviewers for their constructive feedback. The reviewers point out some remaining elements requiring modifications or clarifications in order to validate my manuscript for publication. My manuscript was thoroughly reviewed and updated according to these pertinent remarks. I would like to illustrate these modifications and address those discussion points in my response below.

Point 1: The manuscript presents extremely relevant data for the organization of health actions; however, some aspects need to be better described. The title of the manuscript leads us to the understanding of the services offered in dental clinics, however, when reading the text there is little focus specifically on this service, since the text is more centered on doctor and medical services. The suggestion is that the text be described about health services in general, and specifically about dental clinics and the dentist. The research was carried out on dental clinics and dentists, however, there is little presence of these terms throughout the text, which allows for a confused reading of the text by over-mentioning the terms doctor and medical services.

Response 1: P.3

In general, in order to receive care in Korea, primary and secondary medical institutions must be visited first, and in case of major ailment patients, a medical referral form is issued and care can be received at tertiary medical institutions. However, most dental clinics are composed of primary and secondary medical institutions, and primary medical institutions can receive treatment at the level of tertiary medical institutions. In addition, most of the dental clinics tend to have outpatients and relatively few patients with severe diseases. Therefore, it is necessary to increase the patient's reuse intention through patient-centered communication because the patient can choose a dental clinics to receive treatment [3].

Point 2: The abstract needs to present the context of dental clinics and not medical services, it is important to highlight the object of the study. It is necessary to review the verbs and present them in the past. The last sentence of the methods is incomplete. Replace p=0.000 with p<0.001, considering that there is no statistical significance equal to zero. In the statistical packages, when checking the output, it is possible to verify the exact significance.

Response 2: P.2

Abstract

Introduction

Recent changes in the medical paradigm are highlighting the importance of patient-centered communication. However, due to the lack of awareness of dental clinics and of competence in medical personnel, the quality of medical services regarding communication between doctors and patients has not improved. This study analyzes the impact of health communication and medical service quality, service value, and patient satisfaction on revisiting intention for dental clinics.

Methods

The study subjects were outpatients who were treated at 10 dental clinics in Seoul. The research data were collected using a questionnaire visited the dental clinics from December 1 to December 30, 2016. A total of 600 questionnaires were distributed to 10 dental clinics, 60 copies each, and 570 valid questionnaires were used for analysis. In this study, the structural influence of factors was determined using structural equation modeling. 

Results

The factors influencing service value were reliability (β = 0.364, p < 0.001), expertise (β = 0.319, p < 0.001), communication by a doctor (β = 0.224, p < 0.001) and tangibles (β = 0.136, p < 0.05). In addition, the factors influencing patient satisfaction were in the order of reliability (β = 0.258, p < 0.001), tangibility (β = 0.192, p < 0.001), communication by a doctor (β = 0.163, p < 0.001) and Expertise (β = 0.122, p < 0.01). On the other hand, service value (β = 0.438, p < 0.001) had a positive effect on patient satisfaction, and patient satisfaction (β = 0.383, p < 0.001) was found to influence dental clinics the reuse intention.

Point 3: Introduction

The first paragraph has no reference. Review throughout the text to prioritize the use of the terms: dentists and dental clinics. It is necessary to review the null hypothesis presented since this hypothesis generally states that there is no relationship between the studied phenomena. The objectives need to be better described, as there is duplication in the presentation.

Response 3: P.3

Medical services are changing from a disease-centered model to a patient-centered model. In the existing disease-centered model, all the decision-making concerning patient care was conducted based on the expertise of doctors and other medical personnel, but in the patient-centered model, the patient actively participates in their treatment process and their needs and preferences are reflected in care-related decision making [1, 2].

To this end, the null hypothesis was established that health communication and medical service quality do not affect the reuse intention of dental clinics by mediating service value and patient satisfaction.

Point 4: Methods

The text does not make it clear what the inclusion criteria were. Were people under 18 included?

The data analysis section needs to be reviewed carefully. It is necessary to describe in detail the analyzes carried out, as well as the criteria used for each type of analysis. 

Response 4: P.6

We focused on patients waiting to make a payment or to receive their prescription after receiving treatment as an outpatient over 13 years old in the dental clinic. Since oral care can lead to chronic diseases, regular checkups, and prevention are required from adolescence. The quality of dental services felt by patients during their adolescence can lead to adults, which can be an obstacle to continuous visits to dental institutions, so adolescent patients were included in the study.

Point 5: What were the criteria and procedures adopted for the factor analysis? What were the post-tests used to assess the adequacy of the model? What criteria are used?

Response 5: P.7

Second, factor analysis was performed to verify the validity of the questions, while the reliability of the measurement questions was validated using Cronbach's alpha (α). In the case of factor analysis, first, an Exploratory Factor Analysis (EFA) of the Varimax mode orthogonal rotation was performed to examine the factor structure of the questions for measuring variables. Next, a Confirmatory Factor Analysis (CFA) was conducted to confirm whether the derived factor structure was consistent with actual empirical data.

Point 6: The results related to factor analysis were not presented. What were the communalities, the sample adequacy measures, the variance explained by each factor?

Response 6: P.10

An EFA was conducted based on the collected data to examine the factor structure of Forty-eight questions used to measure variables. Factor analysis was carried out by removing the items that hinder the validity and deleting the questions with the lowest commonality. After that, the question with the lowest commonality was deleted and the work of factor analysis was repeated. As a result, the items of expertise of assistant staff, responsiveness of the office or clinic had low commonality and were deleted, and finally, forty-two questions were selected. EFA was again conducted to examine the factor structure of the final selected items. The Karser Meyer Olkin (KMO) = 0.944, Bartlett's test of sphericity test was also significant (χ₂ = 13748.522, P < 0.001), and the data used in the analysis were judged to be suitable for factor analysis. In addition, the Total Variance Explained was 74%.

Point 7: In the instrument used, there are more dimensions than those shown in Table 2. The dimensions “Expertise of assistant staff” and “Responsiveness of the office or clinic” are not listed in the table. The number of items in the Communication by assistant dimension is different on the table and on the instrument.

Response 7: P.10

An EFA was conducted based on the collected data to examine the factor structure of Forty-eight questions used to measure variables. Factor analysis was carried out by removing the items that hinder the validity and deleting the questions with the lowest commonality. After that, the question with the lowest commonality was deleted and the work of factor analysis was repeated. As a result, the items of expertise of assistant staff, responsiveness of the office or clinic had low commonality and were deleted, and finally, forty-two questions were selected. EFA was again conducted to examine the factor structure of the final selected items. The Karser Meyer Olkin (KMO) = 0.944, Bartlett's test of sphericity test was also significant (χ₂ = 13748.522, P < 0.001), and the data used in the analysis were judged to be suitable for factor analysis. In addition, the Total Variance Explained was 74%.

Point 8: The instrument contains 48 items; however, it was presented that the analysis was performed with only 38 items, it is necessary to present the reasons that led to the exclusion of 10 items.

Response 8: P.11

In the CFA, two items on communication by doctors, one on tangibility, and one on accessibility did not exceed the 0.5 standard factor loading criterion. That is, out of the Forty-eight questions used for data collection, six items with poor commonality were removed through EFA, and four items with poor validity were removed through CFA. Therefore, 38 items—excluding ten items—were used for the analysis. Table 3 shows the results of the CFA for the model used in this study.

Point 9: In table 5, it is necessary to review the presentation of the p-values (p=0.000). If the p-value was presented, it is not necessary to use symbols to describe the statistical significance.

Response 9: P.14

Table 5. Research model path coefficients.

Factor Path B β S.E. T p-value

Service value Service ← Communication by doctor 0.215 0.224 0.060 3.600** 0.001

 Service ← Communication by assistant -0.038 -0.037 0.066 -0.580 0.562

 Service ← Expertise 0.321 0.319 0.086 3.748*** 0.001

 Service ← Reliability 0.365 0.364 0.089 4.113*** 0.001

 Service ← Tangibility 0.175 0.136 0.081 2.145** 0.032

 Service ← Accessibility 0.014 0.014 0.052 0.259 0.795

Patient satisfaction Satisfaction ← Communication by doctor 0.140 0.163 0.044 3.211** 0.001

 Satisfaction ← Communication by assistant 0.046 0.050 0.046 0.997 0.319

 Satisfaction ← Expertise 0.110 0.122 0.062 1.778* 0.075

 Satisfaction ← Reliability 0.231 0.258 0.064 3.585*** 0.001

 Satisfaction ← Tangibility 0.220 0.192 0.059 3.729*** 0.001

 Satisfaction ← Accessibility -0.005 -0.005 0.037 -0.130 0.897

 Satisfaction ← Service value 0.397 0.444 0.050 7.941*** 0.001

Reuse intention Reuse ← Patient satisfaction 0.491 0.383 0.087 5.616*** 0.001

 Reuse ← Service value 0.501 0.438 0.078 6.414*** 0.001

*p<0.1, **p<0.05, ***p<0.001, B=Unstandardized coefficients, β=Standardized coefficients, S.E=Standard error, T=t-value

Point 10: In figure 2, the expertise dimension has repeated values, it is necessary to correct it. As this is the figure that presents the final model, I suggest that only the relations that were significant for the composition of the final model be presented.

Response 10: P.15

Fig 2. Final path model.

Point 11: In view of the notes made, it is suggested that all dimensions of the final model be addressed in the discussion. It is necessary to review the use of the terms doctor and medical services.

Response 11: P.15-16

As a result of the study, health communication and quality of medical service influenced the reuse intention of dental clinics by mediating patient satisfaction and service value, thus rejecting the null hypothesis and adopting the alternative hypothesis. The detailed study results are as follows.

Point 12: It is highly recommended that the title, objectives and conclusion are related and that the conclusion responds directly to the proposed objectives.

Response 12: P.18

This study aimed to analyze the influencing factors of the reuse of medical services, employing data from patients visiting dental clinics located in Seoul. The results showed that reliability and communication by doctors affected service value and patient satisfaction, which had an effect on reuse intention. It is considered that the following measures are necessary to increase the satisfaction of patients who have visiting dental clinics and to increase the reuse intention. Dental clinics should provide appropriate medical services to outpatients, which is based on smooth communication between doctors and patients. Additionally, doctors having an attitude of respect toward the patient may affect patient satisfaction. Doctors providing medical treatment information to patients with a friendly and respectful attitude rather than an authoritarian attitude may be an effective strategy for dental medical institutions to attract long-term customers.

I hope that the considerable changes made to my research paper coupled with our above arguments will convince the reviewers in reconsidering our manuscript for publication in your journal. I thank you in advance for your kind and thorough attention in the review of my work.

Best regards,

---

## [Decision Letter · Decision Letter 3]

5 Feb 2021

PONE-D-19-29416R3

Factors affecting revisiting intention for medical services at dental clinics

PLOS ONE

Dear Dr. Lee,

Thank you for submitting your manuscript to PLOS ONE. After careful consideration, we feel that it has merit but still does not fully meet PLOS ONE’s publication criteria as it currently stands. Therefore, we invite you to re-submit a revised version of the manuscript that addresses the points raised during the review process.

Having intensively reviewed your draft, our reviewers again have indicated that your re-submitted draft might be perfectible. All in all, the indicated shortcomings would seem reasonable, and your current version would not seem satisfying. Please note that a final proceeding will be possible with faultless manuscripts only. Moreover, one of our reviewers has asked for more complete statistical explanations. Remember that reproducibility is the cornerstone of scientific advancement, so please ensure to re-submit replicable information and descriptions with your convincingly revised draft.  

Indeed, you should follow the reviewers' comments, to finalize your paper, and to meet both PLOS ONE's quality standards and our readership's expectations. Please note that a non-convincing revision (not considered acceptable with regard to language, reviewers' constructive criticism, content, generalizable outcome, and/or Authors' Guidelines) will lead to outright reject. 

We look forward to receiving your revised manuscript.

Kind regards,

Andrej M Kielbassa

Academic Editor

PLOS ONE

Reviewers' comments:

Reviewer's Responses to Questions

**Comments to the Author**

1. If the authors have adequately addressed your comments raised in a previous round of review and you feel that this manuscript is now acceptable for publication, you may indicate that here to bypass the “Comments to the Author” section, enter your conflict of interest statement in the “Confidential to Editor” section, and submit your "Accept" recommendation.

Reviewer #1: (No Response)

Reviewer #4: (No Response)

2. Is the manuscript technically sound, and do the data support the conclusions?

Reviewer #1: Yes

Reviewer #4: Partly

3. Has the statistical analysis been performed appropriately and rigorously? 

Reviewer #1: Yes

Reviewer #4: No

4. Have the authors made all data underlying the findings in their manuscript fully available?

Reviewer #1: Yes

Reviewer #4: No

5. Is the manuscript presented in an intelligible fashion and written in standard English?

Reviewer #1: Yes

Reviewer #4: No

6. Review Comments to the Author

Reviewer #1: Still, some minor typos are evident with the Reference section, and the latter would seem lacking uniformity. These aspects will be revised with the proofs, so please pay special attention to the proof reading. This revised and re-submitted manuscript would seem ready to proceed.

Reviewer #4: The authors accepted most of the suggestions made and adapted the manuscript accordingly. However, there is still a need for a better description of the statistical analysis plan and presentation of results related to exploratory factor analysis.

In the statistical analysis plan, the criteria used to include the variables in the exploratory factor analysis model were not described. What were the correlation coefficients considered for inclusion in this model? What were the commonality values considered to exclude variables from the model?

In the results, it is strongly recommended that the exploratory factor analysis results be presented in a table with the value of each variable's factor loads in each factor, with the value of the sample adequacy measure, commonality, and percentage of explained variance. All the factors generated must be presented, with their respective identifications and variables included. From these results, it is possible to understand why the factors were created and to know which variables were excluded from the final analysis.

Although I am not a native English speaker, it is strongly suggested to revise the entire text.

7. PLOS authors have the option to publish the peer review history of their article (what does this mean?). If published, this will include your full peer review and any attached files.

Reviewer #1: No

Reviewer #4: No

---

## [Author Response · Author response to Decision Letter 3]

24 Mar 2021

Response to Reviewer Comments

I wish to thank the reviewers for their constructive feedback. The reviewers point out some remaining elements requiring modifications or clarifications in order to validate my manuscript for publication. My manuscript was thoroughly reviewed and updated according to these pertinent remarks. I would like to illustrate these modifications and address those discussion points in my response below.

Point 1: Still, some minor typos are evident with the Reference section, and the latter would seem lacking uniformity. These aspects will be revised with the proofs, so please pay special attention to the proof reading. This revised and re-submitted manuscript would seem ready to proceed.

Response 1: 

Modified it according to the reviewer's advice. Thank you for reviewing my paper so that it can be published.

I hope that the considerable changes made to my research paper coupled with our above arguments will convince the reviewers in reconsidering our manuscript for publication in your journal. I thank you in advance for your kind and thorough attention in the review of my work.

Best regards,

Response to Reviewer Comments

I wish to thank the reviewers for their constructive feedback. The reviewers point out some remaining elements requiring modifications or clarifications in order to validate my manuscript for publication. My manuscript was thoroughly reviewed and updated according to these pertinent remarks. I would like to illustrate these modifications and address those discussion points in my response below.

Point 1: In the statistical analysis plan, the criteria used to include the variables in the exploratory factor analysis model were not described. What were the correlation coefficients considered for inclusion in this model? What were the commonality values considered to exclude variables from the model?

Response 1: P.13 (Line 211-218).

For exploratory factor analysis, the validity of the composition was verified using the principal components analysis (PCA) of the Varimax rotation, and Kaise-Meyer Olkin (KMO) and Barlett sphericity were verified. Variables were selected based on an eigenvalue of 1 or more and factor loading of 0.4 or more for each variable, and Cronbach's Alpha was checked for reliability, and items that lowered reliability were removed through factor analysis and improved to an appropriate level. As a result, six items including the expertise of assistants and responsiveness of the office/clinic had commonality less than 0.4 and were deleted.

Point 2: In the results, it is strongly recommended that the exploratory factor analysis results be presented in a table with the value of each variable's factor loads in each factor, with the value of the sample adequacy measure, commonality, and percentage of explained variance. All the factors generated must be presented, with their respective identifications and variables included. From these results, it is possible to understand why the factors were created and to know which variables were excluded from the final analysis. 

Response 2: P.15-16(Line 223-224).

Table 2. EFA Results

Variable Commonality Component

 1 2 3 4 5 6 7 8 9

Communication by doctor Doctor1 0.698 0.295 

 Doctor2 0.766 0.292 

 Doctor3 0.750 0.251 

 Doctor4 0.769 0.208 

 Doctor5 0.794 0.184 

 Doctor6 0.749 0.182 

 Doctor7 0.766 0.136 

Communication by assistant Assistant1 0.755 0.818 

 Assistant2 0.828 0.798 

 Assistant3 0.814 0.791 

 Assistant4 0.808 0.751 

 Assistant5 0.823 0.732 

Expertise Expertise1 0.672 0.695 

 Expertise2 0.754 0.661 

 Expertise3 0.773 0.603 

Expertise of assistant staff Expertise of assistant1 0.312 

 Expertise of assistant2 0.339 

 Expertise of assistant3 0.392 

Reliability Reliability1 0.789 0.606 

 Reliability2 0.785 0.571 

 Reliability3 0.725 0.566 

 Reliability4 0.719 0.525 

responsiveness Responsiveness1 0.234 

 Responsiveness2 0.351 

 Responsiveness3 0.256 

Tangibility Tangibility1 0.692 0.847 

 Tangibility2 0.692 0.830 

 Tangibility3 0.679 0.806 

 Tangibility4 0.814 0.758 

 Tangibility5 0.809 0.646 

 Tangibility6 0.790 0.495 

Accessibility Accessibility1 0.804 0.817 

 Accessibility2 0.814 0.786 

 Accessibility3 0.721 0.775 

 Accessibility4 0.757 0.673 

Patient satisfaction Patient satisfaction1 0.810 0.655 

 Patient satisfaction2 0.825 0.640 

 Patient satisfaction3 0.846 0.638 

 Patient satisfaction4 0.798 0.568 

Service value Service value1 0.855 0.816 

 Service value2 0.850 0.800 

 Service value3 0.799 0.797 

 Service value4 0.773 0.785 

 Service value5 0.795 0.726 

Revisit intention Revisit intention1 0.882 0.685

 Revisit intention2 0.915 0.631

 Revisit intention3 0.906 0.626

 Revisit intention4 0.899 0.593

Eigenvalue 21.08 3.29 2.65 1.96 1.83 1.41 1.25 1.08 1.02

Explained variance (%) 16.2 11.13 9.83 9.75 9.53 6.15 5.03 3.49 2.89

Total explained variance (%) 16.26 27.39 37.21 46.96 56.54 62.69 67.71 71.21 74.10

Point 3: Although I am not a native English speaker, it is strongly suggested to revise the entire text.

Response 3: 

Modified it according to the reviewer's advice. Thank you for reviewing my paper so that it can be published.

I hope that the considerable changes made to my research paper coupled with our above arguments will convince the reviewers in reconsidering our manuscript for publication in your journal. I thank you in advance for your kind and thorough attention in the review of my work.

Best regards,

---

## [Decision Letter · Decision Letter 4]

12 Apr 2021

Factors affecting revisit intention for medical services at dental clinics

PONE-D-19-29416R4

Dear Dr. Lee,

congratulations and compliments, we’re pleased to inform you that your manuscript has been judged scientifically suitable for publication and will be formally accepted for publication once it meets all outstanding technical requirements.

Again, please accept our congratulations, kind regards, and stay healthy,

Prof. Dr. med. dent. Dr. h. c. Andrej M Kielbassa

Academic Editor

PLOS ONE

Additional Editor Comments (optional):

Reviewers' comments:

Reviewer's Responses to Questions

**Comments to the Author**

1. If the authors have adequately addressed your comments raised in a previous round of review and you feel that this manuscript is now acceptable for publication, you may indicate that here to bypass the “Comments to the Author” section, enter your conflict of interest statement in the “Confidential to Editor” section, and submit your "Accept" recommendation.

Reviewer #1: All comments have been addressed

Reviewer #4: All comments have been addressed

2. Is the manuscript technically sound, and do the data support the conclusions?

Reviewer #1: Yes

Reviewer #4: Yes

3. Has the statistical analysis been performed appropriately and rigorously? 

Reviewer #1: (No Response)

Reviewer #4: Yes

4. Have the authors made all data underlying the findings in their manuscript fully available?

Reviewer #1: Yes

Reviewer #4: (No Response)

5. Is the manuscript presented in an intelligible fashion and written in standard English?

Reviewer #1: Yes

Reviewer #4: (No Response)

6. Review Comments to the Author

Reviewer #1: Revisions would seem satisfying, and paper is ready to proceed. Congrats and compliments, and stay healthy!

Reviewer #4: The authors present themes of extreme relevance to the organization of health services.

The manuscript has clarity and objectivity.

7. PLOS authors have the option to publish the peer review history of their article (what does this mean?). If published, this will include your full peer review and any attached files.

Reviewer #1: No

Reviewer #4: No

---

## [Editor Report · Acceptance letter]

23 Apr 2021

PONE-D-19-29416R4 

Factors affecting revisit intention for medical services at dental clinics 

Dear Dr. Lee:

I'm pleased to inform you that your manuscript has been deemed suitable for publication in PLOS ONE. Congratulations! Your manuscript is now with our production department. 

Kind regards, 

on behalf of

Prof. Dr. med. dent. Dr. h. c. Andrej M Kielbassa 

Academic Editor

PLOS ONE